# Effectiveness of health voucher scheme and micro-health insurance scheme to support the poor and extreme poor in selected urban areas of Bangladesh: An assessment using a mixed-method approach

**Sayem Ahmed**[1,2,3]*, **Md. Zahid Hasan**[4], **Nausad Ali**[4,5], **Mohammad Wahid Ahmed**[4], **Emranul Haq**[6], **Sadia Shabnam**[7], **Morseda Chowdhury**[7], **Breda Gahan**[8], **Christine Bousquet**[8], **Jahangir A. M. Khan**[3,9], **Ziaul Islam**[4]

1 Mathematical Modelling Group, Oxford University Clinical Research Unit (OUCRU), Ho Chi Minh City, Vietnam, 2 Centre for Tropical Medicine and Global Health, Nuffield Department of Medicine, University of Oxford, Oxford, United Kingdom, 3 Liverpool School of Tropical Medicine (LSTM), Liverpool, United Kingdom, 4 International Centre for Diarrhoeal Disease Research, Bangladesh (icddr,b), Dhaka, Bangladesh, 5 Bangladesh Institute of Development Studies (BIDS), Agargaon, Dhaka, Bangladesh, 6 Concern Worldwide, Dhaka, Bangladesh, 7 BRAC, Dhaka, Bangladesh, 8 Concern Worldwide, Dublin, Ireland, 9 Health Economics and Policy Unit, School of Public Health and Community Medicine, Sahlgrenska Academy, University of Gothenburg, Gothenburg, Sweden

* sayem.ahmed@ndm.ox.ac.uk

## Abstract

### Background

National healthcare financing strategy recommends tax-based equity funds and insurance schemes for the poor and extreme poor living in urban slums and pavements as the majority of these population utilise informal providers resulting in adverse health effects and financial hardship. We assessed the effect of a health voucher scheme (HVS) and micro-health insurance (MHI) scheme on healthcare utilisation and out-of-pocket (OOP) payments and the cost of implementing such schemes.

### Methods

HVS and MHI schemes were implemented by Concern Worldwide through selected NGO health centres, referral hospitals, and private healthcare facilities in three City Corporations of Bangladesh from December 2016 to March 2020. A household survey with 1,294 enrolees, key-informant interviews, focus group discussions, consultative meetings, and document reviews were conducted for extracting data on healthcare utilisation, OOP payments, views of enrolees, and suggestions of implementers, and costs of services at the point of care.

### Results

Healthcare utilisation including maternal, neonatal and child health (MNCH) services, particularly from medically trained providers, was higher and OOP payments were lower among

**Data Availability Statement:** All relevant data are within the paper and its Supporting information files.

**Funding:** icddr,b received funding for this research study from the Concern Worldwide Bangladesh. The funding was awarded to Ziaul Islam under grant number GR-01812. icddr,b acknowledges with gratitude the commitment of Concern Worldwide Bangladesh to its research efforts. Concern Worldwide was funded by the European Union (EU) for the overall project.

**Competing interests:** I have read the journal's policy and the authors of this manuscript have the following competing interests: Sayem Ahmed – none declared Md. Zahid Hasan – none declared Nausad Ali – none declared Mohammad Wahid Ahmed – none declared Emranul Haq – works at the Concern Worldwide Bangladesh Sadia Shabnam – works at the scheme implementing organization, BRAC Morseda Chowdhury – works at the scheme implementing organization, BRAC Breda Gahan – works at the funding organization, Concern Worldwide Christine Bousquet – works at the funding organization, Concern Worldwide Jahangir A. M. Khan – none declared Ziaul Islam – none declared This does not alter our adherence to PLOS ONE policies on sharing data and materials.

the scheme enrolees compared to corresponding population groups in general. The beneficiaries were happy with their access to healthcare, especially for MNCH services, and their perceived quality of care was fair enough. They, however, suggested expanding the benefits package, supported by an additional workforce. The cost per beneficiary household for providing services per year was €32 in HVS and €15 in MHI scheme.

## Conclusion

HVS and MHI schemes enabled higher healthcare utilisation at lower OOP payments among the enrolees, who were happy with their access to healthcare, particularly for MNCH services. However, they suggested a larger benefits package in future. The provider's costs of the schemes were reasonable; however, there are potentials of cost containment by purchasing the health services for their beneficiaries in a competitive basis from the market. Scaling up such schemes addressing the drawback would contribute to achieving universal health coverage.

## Introduction

In 2019, for the first time in history, around half of the population in Asia became urban residents accounting for 54% of the world's urban population [1]. In Bangladesh, 47.2% of the total urban population live in slums according to The World Bank [2]. The slum settlements have proliferated with a recent census counting approximately 14,000 slum settlements across the country [3, 4]. The slum populations are mostly internally displaced from rural to urban areas in search of better livelihood. Natural disasters like river erosion and floods largely affect their homestead and cultivable lands fisheries and poultry pushing them into poverty. Despite making remarkable progress in several health indicators, Bangladesh is facing a daunting challenge in providing health care to the urban poor and ultra-poor living in the slums and pavements [5].

The slum populations struggle with access to basic amenities like proper housing, safe water, sanitation, electricity and health care which creates a vicious cycle of infections, malnutrition, and poor health. Diseases such as tuberculosis, dengue fever, and hepatitis B have re-emerged and are more widespread in the urban slums than in rural areas [6–8]. Many women die in slums during pregnancy and delivery and the mortality of children younger than 5 years in slums is almost double than that of in rural areas [9]. Two-thirds of these deaths could be avoided if timely appropriate services were available [5]. Evidence showed that 82.4% of slum dwellers received health care from informal providers [10, 11]. Healthcare utilization from informal providers can have adverse effects on health because of non-guideline-based treatment and overuse of drugs [12, 13].

It was further observed that irrespective of locations (urban or rural), inequality in health benefits (cost of health benefits) was severely pronounced (concentration index = 0.229) in the health system in Bangladesh in favour of richer people and even more pronounced when considering health services consumed from private for-profit providers (concentration index = 0.237) [14]. This study found that through decomposing the health benefits by public, private, and NGO provisions, 95.6% of the inequalities in the health system were explained by private health service provision.

A hub of health economic literature demonstrates that the reliance on out-of-pocket (OOP) expenditure for health services lead to a catastrophic burden for many households in Asia [15]. In Bangladesh, 67.0% of the total healthcare expenditure is borne through OOP payments of households [16]. Due to such payments, 14.2% of households face catastrophic health expenditure (CHE) with almost 5 million people falling into poverty in 2010, and similar outcomes were observed in other years [15, 17, 18]. Particularly, 10.6% of the poorest urban households face CHE due to OOP healthcare payments, compared to 6.8% of the richest households [14]. It was also observed that among those who sought healthcare, 41.6% of them utilized services from informal providers (village doctor, drug-sellers) and traditional providers (faith-based healers) [19], which in many cases resulted in over-utilisation of drugs and adverse effects [20–23]. In addition, for low-income people appropriate healthcare was often unaffordable, especially when they relied on OOP payments for accessing such care.

For mitigating these consequences of dependence on OOP payments (inaccessibility to adequate healthcare, falling into poverty etc.), risk-pooling mechanisms are recommended for financing healthcare which further helps towards achieving universal health coverage (UHC). International development partners including the World Health Organization and the World Bank prioritized promoting UHC as a major goal for health system reform in developing countries, with UHC strongly addressed in target 3.8 of the Sustainable Development Goals of the United Nations [24]. With such a large and growing number of extreme poor (living at 60–70 cents per person per day) in big urban areas, achieving UHC in Bangladesh and similar countries is a big challenge. The first-ever healthcare financing strategy of Bangladesh identified poor people (living below 1.25 USD but above 70 cents per person per day) and informal sector workers as the major target population for inclusion in a sustainable healthcare financing system [25]. Community-based and Micro Health Insurance (MHI) schemes have been recommended as a method for financing healthcare of informal workers and the tax-funded scheme for people below the poverty line. Efforts have been undertaken by an international NGO 'Concern Worldwide' in collaboration with the European Union, which were in alignment with the healthcare financing strategy of Bangladesh [25].

## Description of the schemes

In line with the current strategic priorities of the Ministry of Health and Family Welfare (MoHFW) of the Government of Bangladesh, Concern Worldwide implemented a project under a European Union (EU) grant entitled "Improving health and nutrition status of urban extreme poor in Bangladesh through sustainable health service provision" from November 2016 to March 2020. The primary focus of the project was to ensure access and utilisation of better health care services to the urban poor and extreme poor populations. The project aimed to test several approaches and models to 'reach the unreached' in urban settings, and to determine how a sustainable health care delivery system could be established. Prior to these other projects of Concern Worldwide focussed on smaller urban contexts, with the big cities including Dhaka or Chattogram excluded due to the existence of a complex health system and large population size [26]. Two separate financing mechanisms were implemented: Health Voucher Scheme (HVS) in each of Dhaka North City Corporation, Dhaka South City Corporation and Chattogram City Corporation, and an MHI scheme in Dhaka South City Corporation area.

The HVS and MHI scheme was implemented mainly through the existing NGO health facilities e.g. BRAC's *Manoshi* (birthing hut) centres, Marie Stopes clinic, Smiling Sun Clinic, public sector referral hospitals and contracted private facilities to support the scheme enrolees. The HVS needed no payment for enrolment. The MHI scheme enrolled households at different yearly premium levels, i.e., 500 extreme poor households got cards free of cost, 2,400

households paid 350 BDT yearly and 1,100 households paid 550 BDT yearly for accessing the benefits. The MHI scheme was administered by the *Progoti Insurance Limited* on a contractual basis.

Benefit packages of HVS and MHI are presented in Table 1. HVS focused on primary healthcare services which include consultation, medicines and investigations. Notably, all types of maternal, neonatal and child health (MNCH) services are included along with communicable diseases for diagnosis and treatment, whilst family planning, reproductive health, adolescent healthcare and non-communicable diseases were included for consultation only.

The benefit package of the MHI scheme was larger, additionally including inpatient care, specialist consultations, operation costs, ICU/CCU, post-operative care, and C-section (Table 1).

The growing extreme poor population in big urban areas (like metropolitan cities) and the large informal sector of the economy are Bangladesh's two major challenges for achieving UHC. The healthcare financing strategy of Bangladesh suggested that the financing of healthcare of extreme poor should be managed by government's tax revenue and of the low-income informal sector workers by community-based health insurance [25]. This current research assessed the potential effectiveness of a health voucher scheme and micro-health insurance scheme implemented in Dhaka and Chattogram City Corporation areas, on healthcare utilisation and OOP payments and also studied the views and experiences of the clients and scheme implementers as well as the programme cost of these two schemes. The study findings are expected to be useful for policymakers and future programme implementers for scaling-up

**Table 1. Benefits package for enrolees of HVS scheme.**

| HVS scheme | MHI scheme |
|---|---|
| **Treatment and consultation**:<br>• Doctor consultation<br>• Antenatal care<br>• Normal delivery<br>• C-Section (if doctor refer)<br>• Comprehensive antenatal care with BMI<br>• Postnatal care<br>• Neonatal care<br>• Child health care<br>• Communicable diseases<br>• General diseases<br>• Medicine | **Overall benefit package for MHI**<br>• Outdoor services<br>• Indoor services<br>• Maternity services<br>• Consultation with specialist physician<br>• Diagnostic cost<br>• Essential medicines<br>• Operation cost<br>• ICU/CCU<br>• Post-operative care<br>• Blood transfusion<br>• Dressing<br>• Nebulization<br>• Ultrasongram, ECG, X-ray<br>• Normal delivery |
| **Only for consultation**:<br>• Family planning<br>• Reproductive healthcare<br>• Adolescent healthcare<br>• Non-communicable diseases | |
| **Investigations**:<br>• Pregnancy check up<br>• Complete blood count (CBC)/ Hemoglobin (Hgb)<br>• Blood grouping<br>• Urine routine examination (URE)Random blood sugar (RBS) Ultrasonography<br>• Venereal Disease Research Laboratory (VDRL)Hepatitis B | |

Source: Project database.

micro health insurance scheme and health voucher scheme for the urban poor and extreme poor.

## Materials and method

### Study design

This study had three major components, i.e., a cross-sectional household survey, qualitative investigations, and provider cost analysis. A top-down costing approach was adopted for provider's cost analysis for implementing HVS and MHI pilot schemes in Dhaka North, Dhaka South and Chattogram City Corporations and MHI scheme in Dhaka South City Corporation area. It was observed that the large majority of the survey respondents were female at the age of 20–30 years in all three schemes.

Focus group discussion (FGD) and key informant interview (KII) were carried out at the study sites with purposively selected participants to capture the views of the scheme beneficiaries and service providers while service utilization data and cost data were collected from and validated with the project database, BRAC sources, and record review. In absence of a baseline, control or comparison group, we compared the health service utilisation and OOP by the scheme enrolees with the latest Urban Health Survey of Bangladesh [11] and two other studies i.e., Labor Association for Social Protection (LASP) 2012 and Diabetic Association of Bangladesh's (BADAS) Health Insurance Scheme for Ready Made Garment (RMG) Workers 2015. The latter two studies had similar applications as HVS and MHI. Monetary values shown in previous studies were adjusted for inflation for comparison.

### Population and sample size

The total population of HVS and MHI scheme covered 33,126 and 4,000 households respectively. The populations of each scheme were distributed into different geographic areas as presented in Table 2.

Our sampling covered the City Corporation areas of Dhaka (North and South) and Chattogram City Corporation in order to capture knowledge differentials due to geographic location.

**Table 2. Number of cards from Health Voucher Scheme (HVS) and Micro Health Insurance (MHI) schemes distributed to households in different areas in Bangladesh.**

| HVS scheme | | MHI Scheme | |
|---|---|---|---|
| **Area** | **Beneficiary households (Card holders)** | **Area** | **Beneficiary households (Card holders)** |
| Chattogram City Corporation | 7,994 | Kamrangirchar, Dhaka | 1,402 |
| PDC in Chattogram City Corporation | 501 | Rasulpur, Dhaka | 1,504 |
| **Total beneficiary households in** Chattogram City Corporation | **8,495** | Rayerbazar, Dhaka | 496 |
| Dhaka North City Corporation | 12,427 | Islambag, Dhaka | 598 |
| Dhaka South City Corporation | 7,485 | | |
| Pavement Dwellers Centre in Dhaka | 1,717 | | |
| **Total beneficiary households in Dhaka** | **21,629** | | |
| **Mymensingh**[*] | **3,002** | | |
| **Total** | **33,126** | | **4,000** |

Source: Project database.

[*]Not included in this study.

Sample size calculation was based on a previous study on community-based health insurance in Bangladesh that found the health service utilisation rates in the intervention (scheme enrolees) and their matched-control were 16.4% and 12.27% respectively [27]. Considering the same expectation level of increase in healthcare utilisation in the current HVS and MHI scheme, with 90% power and 95% confidence interval, a minimum sample size of 604 households/cards was required. With a 10% non-response rate, the minimum sample size became 665.

Due to variations in the number of cardholders in different locations under different schemes, we applied the probability proportional to size (PPS) sampling technique among three City Corporations for HVS and MHI scheme under investigation which was estimated at 1208 prior to the survey. Finally, we have interviewed a total of N = 1294 cardholders (759 households for HVS in Dhaka North City Corporation and Dhaka South City Corporation, 325 households for HVS in Chattogram City Corporation and 210 for MHI scheme in Dhaka South City Corporation). Based on "yes' consent to participate and availability of the interviewees we interviewed the sample cardholders/household heads in each area until the target was fulfilled or more.

A total of six FGDs were conducted with purposively selected beneficiaries of the two schemes at Dhaka and Chattogram that led to saturation of information (for HVS; 4 FGDs with two in Chattogram City Corporation and two in Dhaka South City Corporation and Dhaka North City Corporation and for MHI; 2 FGDs at Rasulbagh and Islampur of Dhaka city). Based on availability and exposure to the schemes 8–10 cardholders were purposefully selected for each FGD. Eight KIIs were conducted with two purposively selected *Shaystho Kormis* (SK/ health workers)two programme managers, and two branch managers from BRAC, and two ward counsellors (elected public representative) under both the schemes using separate thematic guidelines.

## Data collection

Ten data collectors were recruited and trained by icddr,b investigators. These data collectors conducted face-to-face interviews with the cardholders/household heads during the survey. SK of BRAC assisted the data collectors in finding the location of sample households. A semi-structured questionnaire in *Bangla* was inserted in tab using KoBoToolbox to conduct the interviews with household heads. Qualitative investigations (FGD, KII) were organized and facilitated by trained data collectors. The FGDs and KIIs were conducted by the study investigators and a note-taker took notes of the interviews. All the FGDs and KIIs sessions were audio-recorded with the prior permission of the participants.

Thematic guidelines were developed in Bengali separately for FGDs and KIIs (S1 File). The key themes/areas of interest of FGD and KII included (a) participant's knowledge and experience about the schemes (b) their satisfaction with the services (c) barriers to utilisation and (d) exposure to health education messages. Suggestion/opinion obtained from the participants on the above-mentioned key themes was considered as sub-themes.

The cost data of the HVS and MHI schemes were collected through interviewing BRAC sources, reviewing the project database and other related documents/records using appropriate templates. The questionnaire, guidelines, and checklists were pretested before starting of the final data collection. Data were collected from August to September 2019.

## Programme/Provider cost analysis

The study used a 'top-down' approach for calculating the cost of inputs mobilized for HVS and MHI schemes [28–30]. This costing exercise considered only the implementation related

costs of the schemes, meaning that it captured all inputs used for operating the scheme at the field-level (management and operation) with support from BRAC and Concern Worldwide. Any costs related to the planning of the schemes in the earlier stage and conducting the research was ignored. Information related to inputs and costs were collected from the implementing organization (BRAC) and market values were used whenever applicable. The cost items were classified into "fixed cost" and "variable cost". The costs that didn't vary with the number of beneficiaries were considered as fixed costs e.g. staff training, capital items (laptops, scanner etc.), software, staff salary, BCC materials (poster, leaflets). Costs that varied with the number of beneficiaries were considered as variable costs (e.g., incentive of SKs, reimbursement of claims/treatment cost, diagnostic tests, meetings, and periodic social mobilization activities etc.). To calculate the annual cost for implementing the schemes, we annuitized the costs of capital items (e.g., software development, scanner, laptop, computers) using 3% discounting rate [30]. Costs incurred upon different years were adjusted for inflation at 2019 price level.

The total programme costs were calculated summing up fixed cost and variable cost. The average cost per beneficiary household per year was estimated by dividing the total programme cost by the number of beneficiary households by type of schemes/programmes separately. We didn't consider health facilities cost (i.e. building, furniture, utilities) in the analysis because all treatment costs for study beneficiaries were provided by contracted organizations against the payments for the schemes.

## Data analysis

We estimated total costs and cost per enrolee (unit cost) in the HVS and MHI scheme. Along with mean values, the standard deviation was calculated for understanding the variation of the cost estimate. Total and unit costs were separated into fixed and variable costs. Such disaggregation could be useful for modelling the costs of the schemes considering staff costs, training costs, meeting, social mobilization costs as well as treatment costs for scaling-up.

Descriptive analyses and statistical inference tests were performed. The survey dataset is available in S2 File. The rate of health service utilisation among scheme beneficiaries and their OOP payment for accessing such care were estimated. Survey data on antenatal care (ANC) and postnatal care (PNC) were validated with programme data as deemed necessary. While estimating OOP expenditure, the outliers in the reported OOP expenditure were statistically identified using 'hadimvo' command in statistical package STATA version 14 and removed [31]. We have presented estimates of OOP payments both with and without outliers.

Two types of multiple regression analyses, e.g., log-linear and logistic, were conducted for identifying factors which influenced the service utilisation and OOP payments of scheme enrolees in HVS and MHI scheme. The household demographic and socioeconomic characteristics were used as control variables in the regression analysis for assessing the adjusted effects of HVS and MHI scheme enrolees on healthcare utilisation and OOP payments. In the first model, we used a natural log form of the OOP expenditure $ln (Y_1)$ as dependent variable. The model was specified as follows;

$$ln (Y_{1i}) = \beta_0 + \beta_1 X_{1i} + \beta_2 X_{2i} + \beta_3 X_{3i} + \cdots + \varepsilon_i \ldots \ldots \ldots \ldots \quad (1)$$

Where $\beta_0$ is a constant, $X_1 X_2, X_3, \ldots$ denote control variables, $\beta_1, \beta_2, \beta_3, \ldots$ represent the estimated coefficients, and $\varepsilon_i$ is the random error term of the model. In the second model, we used a binary dependent variable and the model was specified as follows

$$Logit (Y_{ki}) = \theta_0 + \theta_1 X_{1i} + \theta_2 X_{2i} + \theta_3 X_{3i} + \cdots + u_i \ldots \ldots \ldots \ldots \quad (2)$$

**Table 3. Definition of poverty status.**

| Poverty status | HVS and MHI scheme enrolees | National urban population | LASP scheme enrolees |
|---|---|---|---|
| Extreme poor | 1st quintile | 1st quintile | 1st quintile |
| Poor | 2nd to 5th quintile* | 2nd quintile | 2nd to 5th quintile* |

*Since the schemes targeted up to the poor people of the society, the enrolees in 2nd to 5th quintiles were considered to be poor.

Where, $Y_k$ is the dependent variable for utilizing healthcare from medically trained provider (MTP) and coded as binary (0 = No 1 = Yes). $\theta_0$ is a constant, $X_1$ $X_2$, $X_3$, . . . denote control variables, $\theta_1$ $\theta_2$, $\theta_3$ . . . represent the estimated coefficients, and $u_i$ is the random error term of the model.

The findings (utilisation rate and OOP payment) were compared with the health service utilisation and OOP payments reported by similar population groups in the Urban Health Survey [11] and other published reports and articles.

The effects of HVS and MHI schemes on healthcare utilisation and OOP payment were assessed by comparing the utilization rate and average OOP payment between extreme poor (poor) in the reporting schemes and nationally representative urban extreme poor (poor). In a low- and middle-income country (LMIC) like Bangladesh, the households in the poorest wealth quintile are considered as the extreme poor group and those in the 2nd quintile are considered as the poor group. This definition was adapted from a report entitled, 'Addressing Poverty by USAID' [32]. The HVS and MHI schemes targeted the extreme poor and poor segments of the urban populations. For assessing the effects of the schemes on the utilisation and OOP payments of extreme poor, we compared the estimates from the lowest quintile of the scheme enrolees with the lowest quintile of the urban populations of Bangladesh. The enrolees of HVS and MHI schemes, who belonged to any upper quintiles was considered as poor considering the target population of these schemes. Therefore, for estimating the effects of the same schemes on poor people, we compared the utilisation rate and OOP payment between the households in the 2nd to 5th quintile and the national level urban households in the 2nd quintile. For further validation, we compared the healthcare utilisation and OOP payment of the extreme poor (1st quintile among scheme enrolees) and poor (2nd quintile-5th quintile) with the same quintiles of a community-based health insurance scheme enrolees named LASP scheme. This definition of 'extreme poor' and 'poor' in relation to the socioeconomic status of nationally representative urban population are summarized in Table 3.

Thematic analysis was conducted for qualitative data to present the findings in narrative form. For this, transcripts were prepared based on audio recorded sessions/interviews and field notes. The transcriptions were then checked by other investigators for consistencies and errors. Responses were then coded around the above-mentioned (*a priori*) themes and subthemes. Two investigators prepared summary findings and cross-checked them for generating consensus on findings. Finally, triangulation was conducted to validate information derived from different sources.

## Ethical consideration

The study (protocol number PR-19084) was approved by the Institutional Review Board of International Centre for Diarrhoeal Disease Research, Bangladesh (icddr,b). Informed written consent was taken from all interviewees, and confidentiality and anonymity were ensured.

## Results

### Health service utilisation and out-of-pocket payments by scheme beneficiaries

**Characteristics of the beneficiaries.** A total of 3,129 beneficiaries (all household members of a cardholder) was under HVS in Dhaka North, 903 under MHI Dhaka South City Corporations and 1,431 under HVS in Chattogram City Corporation. MHI had a smaller sample with 903 beneficiaries (Table 4). Half of the beneficiaries were at the age of fewer than 20 years in both schemes. The samples were not significantly different by sex of the beneficiaries.

Half of the individual beneficiaries were married in HVS (50.3%) and MHI (49.9%) in Dhaka, while 46 to 48% were unmarried in Dhaka and Chattogram. In all schemes, widowed, divorced and separated constituted a small proportion ranging between 2.1% and 5.7%. Around one- fourth of the beneficiaries (25.3%) were unemployed followed by housewives (approx. 23%) in all schemes. Among the remaining beneficiaries, 11% were labour in HVS in Dhaka and 9% in Chattogram City Corporation (9.1%) while 7% were factory workers in MHI in Dhaka. Fifty-eight percent of the households had 4–5 members in all three schemes. Very few beneficiaries had disability and were members of other NGO/cooperatives. Majority of the beneficiaries had secondary level schooling (9–10 years) with a share of 35.3% in Dhaka HVS scheme, 35.9% in Chattogram City Corporation HVS scheme and 30.7% in Dhaka MHI scheme. Households were classified into five socioeconomic groups using the asset index.

**Health service utilisation.** Table 5 presents the reported illness and disease conditions of the beneficiaries and their healthcare utilisation pattern. Among the HVS beneficiaries, 41.1%, in Dhaka, 37.4% in Chattogram City Corporation and 42% in MHI reported any kind of ill-ness/symptom. Among those who reported such illness/symptom, a large majority (96.6% in HVS in Dhaka, 99.6% in Chattogram City Corporation and 99.2% in MHI in Dhaka) sought healthcare and the majority of them sought care from a medically trained provider with the highest proportion (91%) in Chattogram HVS. Service utilisation from out-patient-depart-ment was quite high in all three sites.

Utilisation of care from informal providers (e.g. quack, homeopath, drug sellers, self/family treatment) were 25.8%, 7.9% and 32.1% respectively among the members of the HVS- Dhaka, HVS- Chattogram and MHI Dhaka.

Fig 1 presents different types of MNCH services utilised by women who delivered a child in the year prior to data collection. Of all childbirths, 60.0% in HVS Dhaka and 72% in HVS Chattogram had a normal delivery, while 73% in MHI Dhaka had normal deliveries. The rest of the pregnant women underwent a C-section for childbirth. It shows that institutional deliv-ery was quite high under both the schemes in all three sites. Corresponding ANC and PNC coverage were also high with an exception of PNC in HVS Dhaka (11%). The proportion of deliveries at scheme contracted other facilities were 72% under HVS Dhaka and Chattogram sites while it was 64% under MHI Dhaka site.

**Out-of-pocket payments.** Fig 2 represents the amount of OOP expenditure (excluding the outliers) of the beneficiaries for utilising healthcare services during the last 6 months prior to data collection. Overall, average OOP spending per beneficiary under HVS Dhaka was 10.9 Euro (12.2 USD, 1050 BDT) 7.1 Euro (7.9 USD, 670 BDT) under HVS Chattogram and 13.8 Euro (15.5 USD, 1301 BDT) under MHI Dhaka (S1 Table). The OOP healthcare payments of the scheme beneficiaries (6 months prior to survey) including outliers are presented in S2 Table. In all three schemes, OOP payment for diagnostic tests was higher followed by OOP payments for medicine. This is indicative that beneficiary households particularly under HVS spent money for doing tests and purchasing medicines that were not available at the pilot

**Table 4. Characteristics of the individual beneficiaries (household members).**

| Characteristics | HVS scheme | | | | MHI scheme | | p-value |
|---|---|---|---|---|---|---|---|
| | Dhaka (N = 3,129) | | Chattogram (N = 1,431) | | Dhaka (N = 903) | | |
| | % | (95% CI) | % | (95% CI) | % | (95% CI) | |
| **Age group (%)** | | | | | | | |
| < 20 years | 48.9 | (47.1–50.6) | 49 | (46.4–51.6) | 50.6 | (47.3–53.9) | 0.029 |
| 20–30 years | 21.8 | (20.3–23.2) | 20.1 | (18.0–22.1) | 22.7 | (20.0–25.4) | |
| 30–40 years | 15.9 | (14.6–17.2) | 14.5 | (12.7–16.4) | 15 | (12.6–17.3) | |
| 40+ years | 13.4 | (12.2–14.6) | 16.4 | (14.5–18.3) | 11.7 | (9.6–13.8) | |
| **Gender (%)** | | | | | | | |
| Male | 47.2 | (45.5–49.0) | 47.6 | (45.0–50.1) | 50.9 | (47.7–54.2) | 0.135 |
| Female | 52.8 | (51.0–54.5) | 52.4 | (49.9–55.0) | 49.1 | (45.8–52.3) | |
| **Marital status (Married %)** | | | | | | | |
| Married | 50.3 | (48.5–52.0) | 46 | (43.4–48.6) | 49.9 | (46.7–53.2) | <0.000 |
| Unmarried | 45.7 | (43.9–47.4) | 48.3 | (45.7–50.9) | 48 | (44.7–51.2) | |
| Others (Widowed, Divorced and Separated) | 4.1 | (3.4–4.8) | 5.7 | (4.5–6.9) | 2.1 | (1.2–3.0) | |
| **Occupation** | | | | | | | |
| Labour | 10.6 | (9.5–11.7) | 9.1 | (7.6–10.6) | 5.8 | (4.2–7.3) | <0.000 |
| Factory worker | 2.9 | (2.3–3.5) | 3.4 | (2.5–4.4) | 6.6 | (5.0–8.3) | |
| Rickshaw puller | 4 | (3.3–4.7) | 4.5 | (3.5–5.6) | 3 | (1.9–4.1) | |
| Driver | 2.1 | (1.6–2.6) | 1.2 | (0.6–1.8) | 1.6 | (0.7–2.4) | |
| Small business | 3.2 | (2.6–3.9) | 1.3 | (0.7–1.8) | 4.9 | (3.5–6.3) | |
| Service holder | 5.1 | (4.3–5.8) | 3.2 | (2.3–4.1) | 6 | (4.4–7.5) | |
| Student | 18.6 | (17.2–19.9) | 22.9 | (20.8–25.1) | 18.8 | (16.3–21.4) | |
| Unemployed | 25.3 | (23.8–26.8) | 22.2 | (20.1–24.4) | 25 | (22.2–27.9) | |
| Housewife | 22.2 | (20.7–23.7) | 21.6 | (19.5–23.7) | 22.7 | (20.0–25.4) | |
| Other | 6 | (5.2–6.9) | 10.5 | (8.9–12.1) | 5.6 | (4.1–7.2) | |
| **Household size** | | | | | | | |
| 3 persons or less | 22.4 | (20.9–23.9) | 16.6 | (14.7–18.6) | 18.7 | (16.2–21.3) | <0.000 |
| 4–5 persons | 57.9 | (56.2–59.7) | 58.2 | (55.7–60.8) | 57.6 | (54.4–60.8) | |
| 6 persons or more | 19.7 | (18.3–21.0) | 25.2 | (22.9–27.4) | 23.7 | (20.9–26.5) | |
| **Years of schooling group** | | | | | | | |
| No formal education | 21.8 | (20.4–23.2) | 13.6 | (11.8–15.3) | 23.1 | (20.4–25.9) | <0.000 |
| Primary level (years 1–5) | 24.6 | (23.1–26.1) | 29.1 | (26.7–31.4) | 19.7 | (17.1–22.3) | |
| Secondary level (years 9–10) | 35.3 | (33.7–37.0) | 35.9 | (33.5–38.4) | 30.7 | (27.7–33.7) | |
| Higher Secondary level and above (years 11+) | 18.3 | (16.9–19.6) | 21.4 | (19.3–23.5) | 26.5 | (23.6–29.3) | |
| **Disability** | | | | | | | |
| Yes | 1.3 | (0.9–1.7) | 1.7 | (1.0–2.3) | 1.1 | (0.4–1.8) | 0.487 |
| No | 98.7 | (98.3–99.1) | 98.3 | (97.7–99.0) | 98.9 | (98.2–99.6) | |
| **Membership in NGO/cooperatives** | | | | | | | |
| Yes | 2 | (1.6–2.5) | 2.5 | (1.7–3.3) | 6 | (4.4–7.5) | <0.000 |
| No | 98 | (97.5–98.4) | 97.5 | (96.7–98.3) | 94 | (92.5–95.6) | |
| **Assets quintiles** | | | | | | | |
| Poorest | 13.3 | (12.1–14.5) | 38.7 | (36.1–41.2) | 18.4 | (15.9–20.9) | <0.000 |
| 2nd | 20.2 | (18.8–21.6) | 13.1 | (11.3–14.8) | 18.5 | (16.0–21.0) | |
| 3rd | 20.5 | (19.1–21.9) | 17.1 | (15.2–19.1) | 19.8 | (17.2–22.4) | |
| 4th | 20.6 | (19.2–22.0) | 19.0 | (16.9–21.0) | 20.5 | (17.9–23.1) | |
| Richest | 25.4 | (23.9–27.0) | 12.2 | (10.5–13.9) | 22.8 | (20.1–25.5) | |

**Table 5. Pattern of healthcare utilization by type of scheme.**

| | HVS | | | | | | MHI | | | P-value |
|---|---|---|---|---|---|---|---|---|---|---|
| | Dhaka | | | Chattogram | | | Dhaka | | | |
| | N | % | 95% CI | N | % | 95% CI | N | % | 95% CI | |
| **Suffered any illness or symptoms** | | | | | | | | | | |
| No | 1833 | 59 | (57.2–60.6) | 891 | 62.6 | (60.1–65.1) | 523 | 58.3 | (55.0–61.4) | 0.038 |
| Yes | 1278 | 41 | (39.4–42.8) | 532 | 37.4 | (34.9–39.9) | 374 | 41.7 | (38.5–44.9) | |
| **Sought healthcare among those who suffered illness** | | | | | | | | | | |
| No | 43 | 3.4 | (2.5–4.5) | 2 | 0.4 | (0.1–1.5) | 3 | 0.8 | (0.3–2.5) | <0.000 |
| Yes | 1235 | 97 | (95.5–97.5) | 530 | 99.6 | (98.5–99.9) | 371 | 99.2 | (97.5–99.7) | |
| **Sought healthcare from medically trained provider among those who sought healthcare** | | | | | | | | | | |
| No | 351 | 28 | (26.0–31.0) | 46 | 8.7 | (6.6–11.4) | 126 | 34 | (29.3–38.9) | <0.000 |
| Yes | 883 | 72 | (69.0–74.0) | 484 | 91.3 | (88.6–93.4) | 245 | 66 | (61.1–70.7) | |
| **Sought healthcare from informal provider provider/self-treatment** | | | | | | | | | | |
| No | 919 | 74 | (71.7–76.6) | 488 | 92.1 | (89.4–94.0) | 252 | 67.9 | (62.9–72.5) | <0.000 |
| Yes | 319 | 26 | (23.4–28.3) | 42 | 7.9 | (5.9–10.6) | 119 | 32.1 | (27.5–37.0) | |
| **Self-reported illness/symptoms (multiple illness counted)** | | | | | | | | | | |
| MNCH | 320 | 21 | (19.4–23.5) | 45 | 8.4 | (6.3–11.1) | 127 | 29.4 | (25.3–33.9) | <0.000 |
| Communicable disease | 729 | 49 | (46.2–51.3) | 262 | 49.1 | (44.8–53.3) | 213 | 49.3 | (44.6–54.0) | |
| Non- communicable disease | 57 | 3.8 | (2.9–4.9) | 39 | 7.3 | (5.4–9.8) | 23 | 5.3 | (3.6–7.9) | |
| Other condition* | 390 | 26 | (23.9–28.4) | 188 | 35.2 | (31.3–39.4) | 69 | 16 | (12.8–19.7) | |
| **Inpatient care utilized (multiple illness counted)** | | | | | | | | | | |
| No | 1,269 | 88 | (86.2–89.5) | 507 | 95.3 | (93.1–96.8) | 337 | 84.9 | (81.0–88.1) | <0.000 |
| Yes | 174 | 12 | (10.5–13.8) | 25 | 4.7 | (3.2–6.9) | 60 | 15.1 | (11.9–19.0) | |

* Weakness, pain, tumor, eye problem, dental problem etc.

facilities e.g. medicines and laboratory tests/x-ray for injuries, NCDs, eye, ENT or dental problems (see Table 1).

Fig 3 presents the average OOP payment across socioeconomic quintiles in three study areas. Variation in OOP payments was observed across the socioeconomic groups in all the three studied schemes. However, in HVS of Chattogram City Corporation and Dhaka MHI scheme OOP payment gradually increased with the increase of socioeconomic status of the beneficiaries. We found that the first quintile (poorest) in any of the areas had the lowest average OOP payment.

The more specific analysis on OOP payments (excluding outliers) for MNCH services showed (Fig 4) that for ANC, PNC, normal delivery, and C-section under HVS in Dhaka was 6.1 Euro (6.8 USD, 575 BDT), 12.4 Euro (13.8 USD, 1169 BDT), 16.2 Euro (18.2 USD, 1527 BDT), and 37.7 Euro (42.2 USD, 3,555 BDT) respectively (S3 Table). In Chattogram City Corporation, the average OOP spending by enrolees of HVS for ANC and PNC was similar (1.1 Euro, 1.8 USD, 104 BDT). However, these beneficiaries incurred 8.9 Euro (10 USD, 839 BDT) for normal delivery and 47.5 Euro (53.3 USD, 4479 BDT) for C-section. It is notable that under MHI scheme in Dhaka there was no OOP spending for C-section because the scheme covered the cost. We presented the OOP healthcare payments for for MNCH services including outliers in S4 Table.

## Determinants of utilization of MTPs and out-of-pocket payments

**Determinants of utilization of MTPs.** While estimating the HVS in Dhaka, our multiple regression model (Table 6) showed that marital status, self-reported illness and socioeconomic

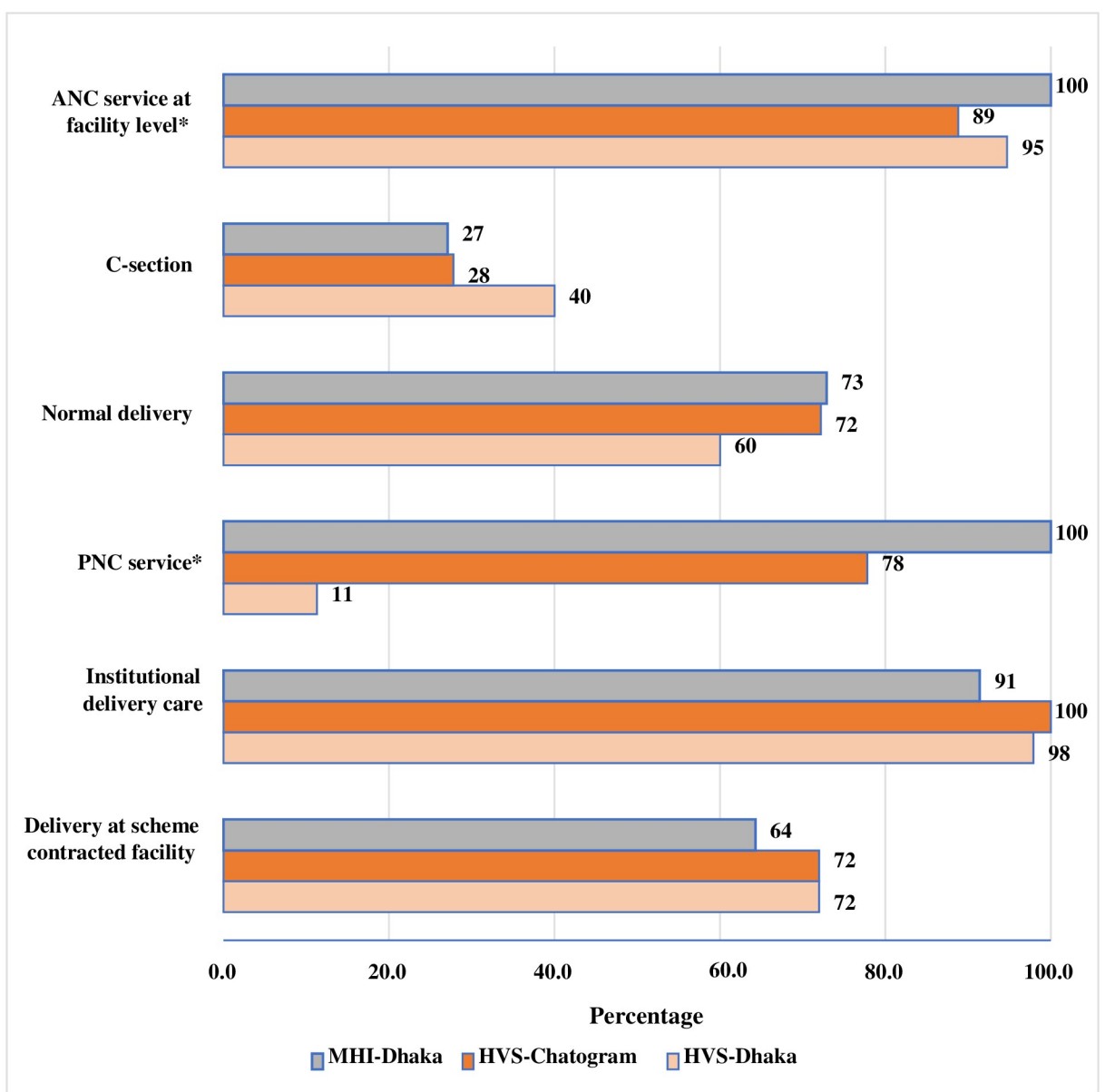

* From program record

**Fig 1. Percentage coverage of MNCH services.**

belonging (asset quintiles) explained the MTP utilisation significantly. Compared to married beneficiaries, the widowed, divorced and separated were 2.6 times more likely to utilize MTPs. Those who reported communicable diseases, non-communicable diseases and other health conditions had significantly lower utilisation than those who reportedly had MNCH related health issues with 96.6%, 89.5% and 85.4% respectively. The 3rd socioeconomic quintile had significantly less utilisation by 43.2% and the richest had 52.3 times higher than the poorest quintile.

The estimation of HVS in Chattogram City Corporation showed that sex, marital status, occupation and socioeconomic status explained the utilisation rate of MTPs significantly. The

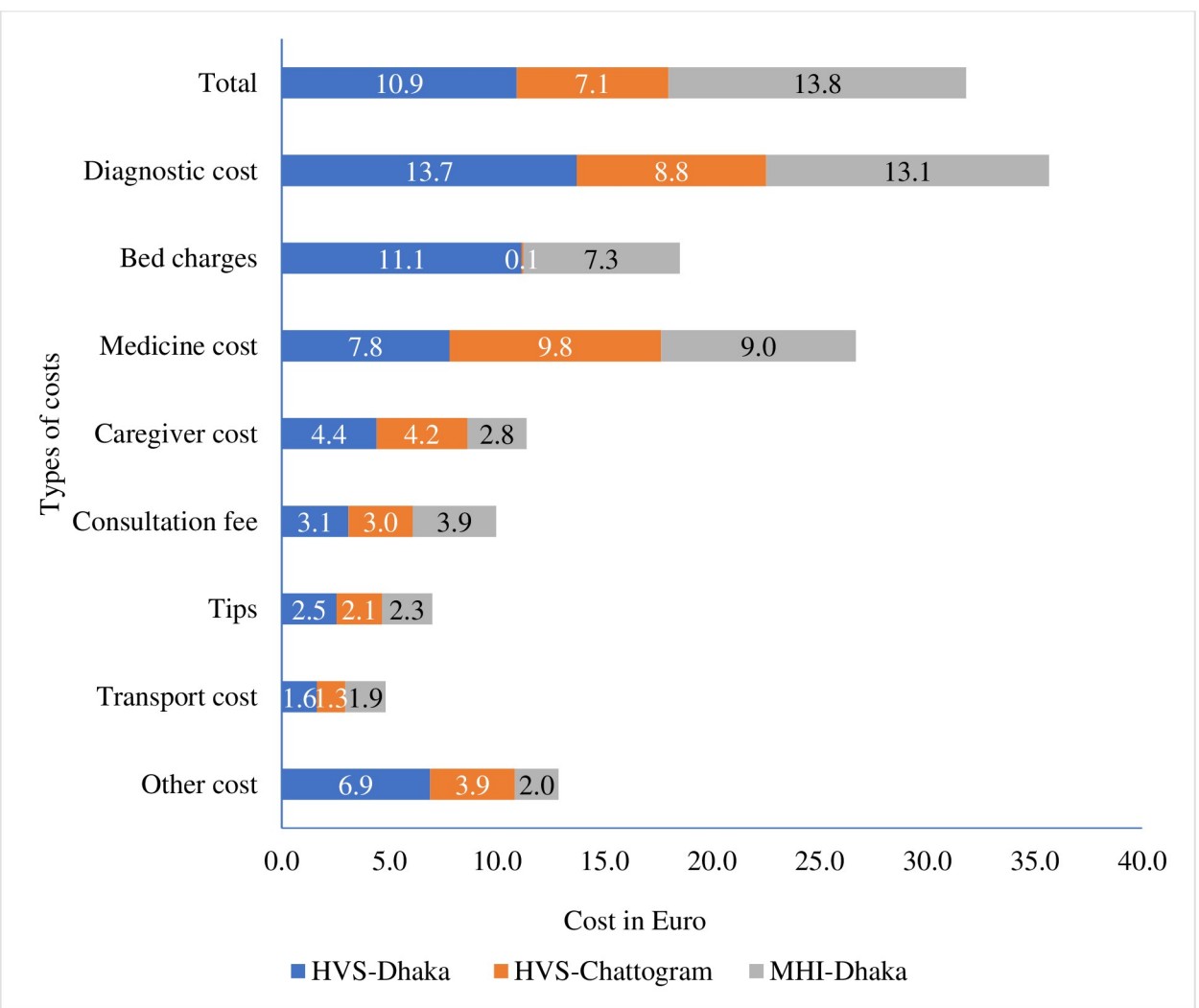

**Fig 2. Overall OOP payments in Euro by scheme beneficiaries (6 months prior to survey).**

male beneficiaries utilized 65.1% less than the females. Unmarried ones utilized almost 7 times higher than the married beneficiaries. Those who were small businessmen utilized the services 92.7% less than the labourers. The beneficiaries in the 4th socioeconomic quintile utilised 89.5% less services than the poorest ones.

In MHI, the MTPs service utilisation was significantly explained by household size and self-reported illness/service. The beneficiaries in households with 4–5 members utilised the services less by 48.9% than those in the households with 3 persons or less. Those who reported communicable, non-communicable and other illness/conditions utilized 97.8%, 87.5% and 91.3% less services, compared to the beneficiaries who reportedly MNCH related health conditions. While comparing MTP utilization in an adjusted model including all schemes, we found that utilization of MTP was 33% lower among the MHI beneficiaries compared to HVS and 78% lower among the beneficiaries from Dhaka compared to the beneficiaries in Chattogram (S5 Table).

**Determinants of out-of-pocket payments.**    The multiple regression models of HVS in Dhaka showed that several factors significantly predicted the OOP expenditure (Table 7). The

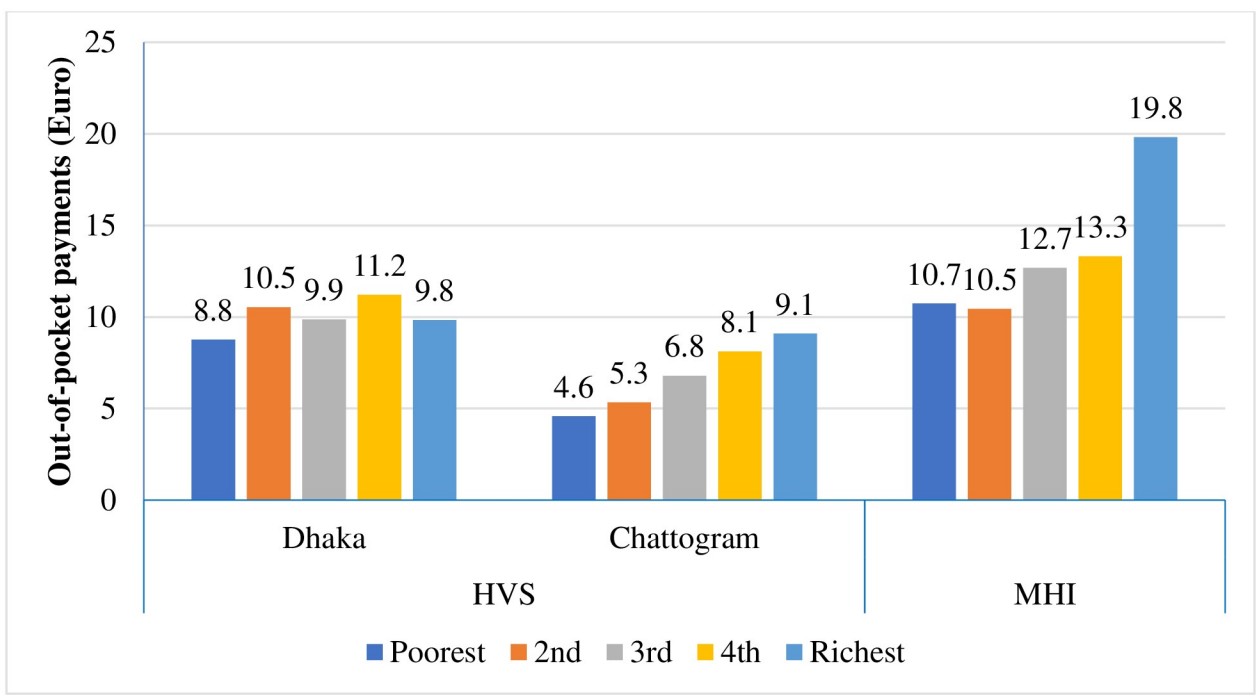

**Fig 3. Average out-of-pocket payment (Euro) across socioeconomic quintiles.**

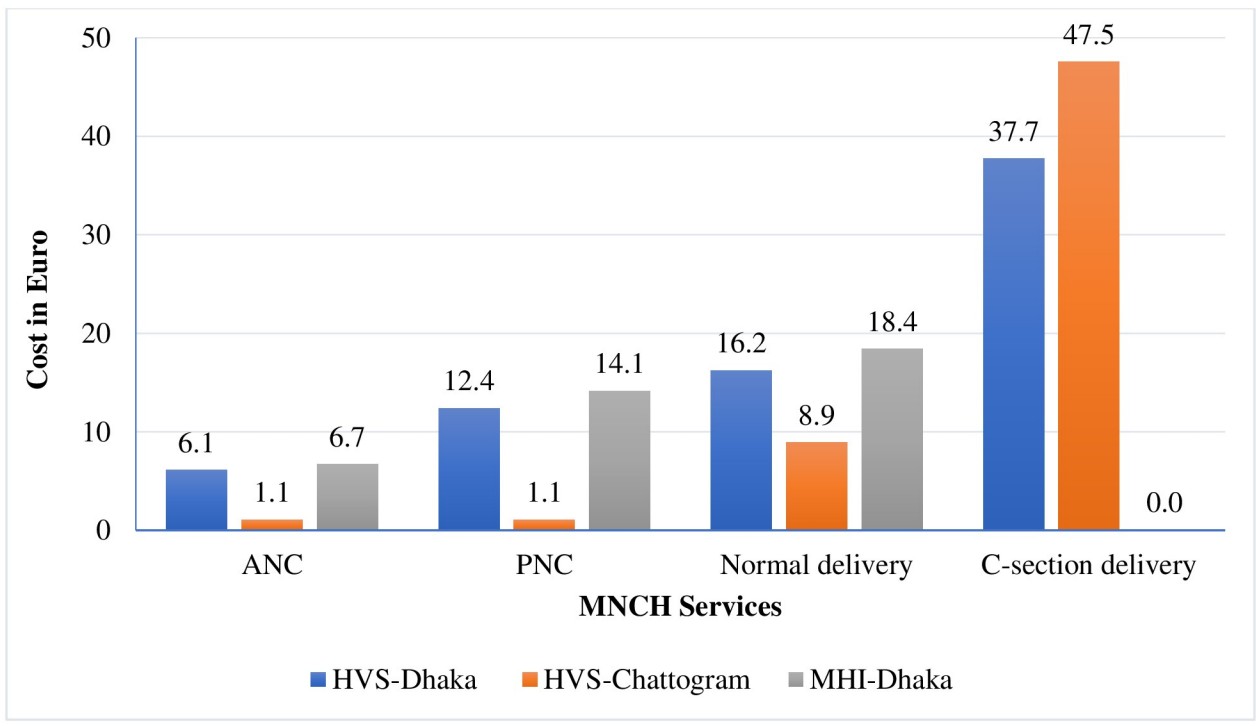

**Fig 4. Out of pocket for MNCH services (six months prior to survey).**

**Table 6. Factors of MTP provider utilization by health schemes.**

| Explanatory variables | HVS | | MHI |
|---|---|---|---|
| | Dhaka | Chattogram | Dhaka |
| | Odds ratio (95% CI) | Odds ratio (95% CI) | Odds ratio (95% CI) |
| **Age group** | | | |
| < 20 years | 1 | 1 | 1 |
| 20–30 years | 0.892 (0.435,1.826) | 3.742 (0.490,28.59) | 1.137 (0.271,4.780) |
| 30–40 years | 1.016 (0.461,2.237) | 2.941 (0.351,24.62) | 1.456 (0.288,7.352) |
| 40+ years | 0.801 (0.350,1.834) | 3.615 (0.417,31.32) | 1.250 (0.241,6.491) |
| **Sex** | | | |
| Female | 1 | 1 | 1 |
| Male | 0.921 (0.677,1.252) | 0.349** (0.145,0.842) | 1.206 (0.658,2.213) |
| **Marital status** | | | |
| Married | 1 | 1 | 1 |
| Unmarried | 1.499 (0.684,3.282) | 6.947* (0.835,57.76) | 0.617 (0.124,3.062) |
| Others (Widowed, Divorced and Separated) | 2.639** (1.140,6.110) | 1.468 (0.264,8.156) | 0.287 (0.0574,1.432) |
| **Occupation** | | | |
| Labour | 1 | 1 | 1 |
| Factory worker | 0.571 (0.195,1.671) | 0.252 (0.0341,1.862) | 0.615 (0.124,3.055) |
| Rickshaw puller | 0.714 (0.297,1.718) | 1.469 (0.128,16.87) | 0.628 (0.0584,6.756) |
| Driver | 0.570 (0.194,1.672) | 0.147 (0.00988,2.198) | 0.509 (0.0566,4.574) |
| Small business | 1.202 (0.467,3.093) | - | 0.592 (0.136,2.585) |
| Service holder | 0.522 (0.237,1.152) | 0.0726*** (0.0127,0.414) | 0.929 (0.188,4.596) |
| Student | 0.789 (0.341,1.827) | 0.205 (0.0232,1.812) | 2.717 (0.528,13.98) |
| Unemployed | 0.577 (0.257,1.294) | 1.885 (0.152,23.41) | 3.005 (0.420,21.51) |
| Housewife | 1.622 (0.870,3.027) | 0.969 (0.209,4.490) | 0.846 (0.218,3.282) |
| Other | 0.847 (0.358,2.004) | 0.809 (0.117,5.619) | 2.519 (0.512,12.40) |
| **Household size** | | | |
| 3 persons or less | 1 | 1 | 1 |
| 4–5 persons | 1.174 (0.862,1.599) | 0.506 (0.159,1.610) | 0.511* (0.256,1.018) |
| 6 persons or more | 1.175 (0.761,1.814) | 0.694 (0.179,2.690) | 0.595 (0.240,1.474) |
| **Years of schooling group** | | | |
| No formal education | 1 | 1 | 1 |
| Up to primary | 0.692 (0.388,1.233) | 2.934 (0.189,45.61) | 0.731 (0.149,3.586) |
| Secondary | 0.783 (0.434,1.412) | 6.504 (0.389,108.8) | 0.479 (0.0996,2.303) |
| Higher secondary and above | 0.943 (0.498,1.786) | 3.927 (0.239,64.40) | 0.777 (0.143,4.222) |
| **Disability** | | | |
| 'No | 1 | 1 | 1 |
| Yes | 1.270 (0.366,4.403) | 0.300 (0.0251,3.591) | 0.0836 (0.00227,3.078) |
| **Membership in NGO/cooperatives** | | | |
| No | 1 | 1 | 1 |
| Yes | 0.950 (0.418,2.161) | 3.257 (0.161,65.91) | 7.456** (1.239,44.86) |
| **Self-reported illness/service** | | | |
| MNCH | 1 | 1 | 1 |
| Communicable disease | 0.0344*** (0.0152,0.0779) | 0.612 (0.262,1.428) | 0.0222*** (0.00687,0.0716) |
| Non-communicable disease | 0.105*** (0.0368,0.300) | 1.404 (0.321,6.146) | 0.125*** (0.0274,0.573) |
| Other condition | 0.146*** (0.0633,0.339) | 1 (1,1) | 0.0869*** (0.0252,0.300) |
| **Assets quintiles** | | | |
| Poorest | 1 | 1 | 1 |

(*Continued*)

**Table 6.** (Continued)

| Explanatory variables | HVS | | MHI |
|---|---|---|---|
| | Dhaka | Chattogram | Dhaka |
| | Odds ratio (95% CI) | Odds ratio (95% CI) | Odds ratio (95% CI) |
| 2nd | 0.822 (0.514,1.315) | 0.857 (0.239,3.070) | 0.583 (0.265,1.281) |
| 3rd | 0.568** (0.356,0.907) | 0.554 (0.190,1.616) | 1.295 (0.588,2.850) |
| 4th | 0.744 (0.467,1.185) | 0.448 (0.164,1.223) | 1.914 (0.783,4.679) |
| Richest | 0.709 (0.446,1.129) | 0.105*** (0.0362,0.306) | 1.399 (0.596,3.284) |
| Constant | 52.34*** (14.55,188.3) | 6.380 (0.213,190.9) | 44.11*** (3.311,587.6) |
| Observations | 1,432 | 478 | 428 |
| Log likelihood | -685.1 | -121.9 | -199.2 |
| Chi-square | 310.8 | 58.94 | 145.6 |
| Degrees of freedom | 29 | 27 | 29 |
| P-value | 0.000 | 0.000 | 0.000 |
| R-square | 0.185 | 0.195 | 0.268 |

\* p<0.10,

\** p<0.05,

\*** p<0.01.

service users aged 40 years and more had 53.5% higher OOP expenditure compared to the youngest (<20 years) ones. The male users spent 62.0% more than the females and widowed, divorced or separated users spent 51.4% more OOP than the married ones. Among the occupational groups, small businessmen and housewives spent 68.2% and 70.0% more than labourers. Beneficiaries from the household with 4–5 members spent 27% less compared to smaller households. The users with primary, secondary, and higher secondary or above level education spent 49.3%, 59.2% and 58.2% less respectively than the users with no formal education. The users with communicable illness spent 94.8% less and with non-communicable illness spent 68.4% more OOP than those who reported MNCH related health conditions. Beneficiaries who had membership in other NGOs/co-operative spent 57.9% more OOP. Those who utilised the services from MTPs spent 66.3% more OOP than those who sought services from non-MTPs.

In the multiple regression model among the users of Chattogram HVS, we found that age, gender, occupation, disability status, types of self-reported illness, and asset quintiles significantly predicted the OOP expenditure. Like Dhaka HVS, users aged 40 years and above in Chattogram HVS spent 118% more compared to the users aged less than twenty years. Male users spent 68.2% more compared to the female and unmarried users spent 126% more OOP compared to the married. Compared to the day-labourer in Chattogram HVS, users who were doing small business and students spent 163% and 110% less OOP. The users with disability had 102.1% more OOP expenditure for healthcare. Like Dhaka HVS, the users with communicable illness spent 170.2% less OOP than those who utilized MNCH care. However, unlike Dhaka HVS, users with non-communicable illnesses spent 119.5% less OOP compared to the users of MNCH care. The users in the 3rd and 4th socioeconomic quintiles had 50.5% and 93.6% more spending respectively than the users in the poorest quintile.

The multiple regression model among the users of MHI scheme found that factors like age, marital status, household size, years of schooling, types of self-reported illness and types of provider utilized significantly predicted the OOP expenditure. The older users had higher

**Table 7. Factors of out-of-pocket (logged) payments by health schemes.**

| Explanatory variables | HVS | | MHI |
|---|---|---|---|
| | Dhaka | Chattogram | Dhaka |
| | Coeff. (95% CI) | Coeff. (95% CI) | Coeff. (95% CI) |
| **Age group** | | | |
| < 20 years | 1 | 1 | 1 |
| 20–30 years | 0.0363 (-0.406,0.478) | 0.238 (-0.718,1.194) | 1.094*** (0.511,1.678) |
| 30–40 years | 0.301 (-0.215,0.818) | 0.863 (-0.264,1.991) | 0.939*** (0.237,1.642) |
| 40+ years | 0.535* (-0.0424,1.112) | 1.180** (0.0280,2.332) | 1.217*** (0.409,2.025) |
| **Sex** | | | |
| Female | 1 | 1 | 1 |
| Male | 0.620*** (0.356,0.885) | 0.682** (0.155,1.209) | -0.00568 (-0.399,0.388) |
| **Marital status** | | | |
| Married | 1 | 1 | 1 |
| Unmarried | -0.0336 (-0.626,0.558) | 1.263** (0.0799,2.446) | 0.998** (0.117,1.879) |
| Others (Widowed, Divorced and Separated) | 0.514* (-0.0905,1.119) | 0.0541 (-0.694,0.802) | -1.389*** (-2.380, -0.398) |
| **Occupation** | | | |
| Labour | 1 | 1 | 1 |
| Factory worker | 0.374 (-0.475,1.224) | -0.714 (-2.013,0.585) | -0.155 (-1.151,0.842) |
| Rickshaw puller | 0.574 (-0.164,1.312) | -0.0931 (-1.678,1.492) | 0.663 (-0.745,2.072) |
| Driver | -0.454 (-1.354,0.446) | -0.309 (-2.216,1.597) | 0.791 (-0.608,2.190) |
| Small business | 0.296 (-0.484,1.076) | -1.630** (-3.000,-0.260) | -0.142 (-1.069,0.784) |
| Service holder | 0.682** (0.0583,1.306) | -0.398 (-1.619,0.822) | -0.295 (-1.274,0.683) |
| Student | 0.456 (-0.232,1.144) | -1.101* (-2.362,0.161) | -0.237 (-1.256,0.782) |
| Unemployed | 0.262 (-0.396,0.919) | -0.330 (-1.495,0.835) | 0.861 (-0.265,1.988) |
| Housewife | 0.700*** (0.201,1.198) | -0.398 (-1.274,0.477) | -0.233 (-1.068,0.602) |
| Other | 0.344 (-0.353,1.042) | -0.751 (-1.783,0.281) | -0.359 (-1.308,0.590) |
| **Household size** | | | |
| 3 persons or less | 1 | 1 | 1 |
| 4–5 persons | -0.270** (-0.510,-0.0287) | -0.0574 (-0.607,0.492) | -0.347* (-0.734,0.0397) |
| 6 persons or more | -0.0498 (-0.386,0.286) | -0.316 (-0.958,0.326) | 0.0918 (-0.415,0.599) |
| **Years of schooling group** | | | |
| No formal education | 1 | 1 | 1 |
| Up to primary | -0.493** (-0.962, -0.0232) | 0.263 (-0.832,1.358) | 0.910* (-0.0389,1.859) |
| Secondary | -0.592** (-1.062,-0.122) | 0.286 (-0.817,1.389) | 0.828* (-0.114,1.771) |
| Higher secondary and above | -0.582** (-1.080,-0.0826) | 0.0928 (-1.000,1.186) | 1.448*** (0.458,2.439) |
| **Disability** | | | |
| No | 1 | 1 | 1 |
| Yes | 0.161 (-1.016,1.338) | 1.021* (-0.177,2.220) | -0.746 (-2.487,0.995) |
| **Membership in NGO/cooperatives** | | | |
| No | 1 | 1 | 1 |
| Yes | 0.579** (0.0183,1.139) | 0.617 (-0.737,1.971) | 0.255 (-0.334,0.843) |
| **Self-reported illness/service** | | | |
| MNCH | 1 | 1 | 1 |
| Communicable disease | -0.948*** (-1.321,-0.574) | -1.702*** (-2.563,-0.842) | -0.315 (-0.899,0.268) |
| Non-communicable disease | 0.684** (0.0957,1.273) | -0.292 (-1.345,0.760) | 0.540 (-0.209,1.290) |
| Other condition | -0.124 (-0.493,0.245) | -1.195*** (-2.027,-0.363) | 0.869*** (0.258,1.481) |
| **Medically trained provider utilization** | | | |
| No | 1 | 1 | 1 |

*(Continued)*

**Table 7.** (Continued)

| Explanatory variables | HVS | | MHI |
| --- | --- | --- | --- |
| | Dhaka | Chattogram | Dhaka |
| | Coeff. (95% CI) | Coeff. (95% CI) | Coeff. (95% CI) |
| Yes | 0.663*** (0.427,0.898) | -0.406 (-1.018,0.205) | 1.118*** (0.766,1.470) |
| **Assets quintiles** | | | |
| Poorest | 1 | 1 | 1 |
| 2nd | 0.0469 (-0.337,0.430) | 0.0551 (-0.560,0.670) | -0.198 (-0.671,0.275) |
| 3rd | 0.117 (-0.260,0.494) | 0.505* (-0.0450,1.055) | -0.147 (-0.618,0.324) |
| 4th | 0.257 (-0.115,0.628) | 0.936*** (0.358,1.514) | -0.151 (-0.649,0.348) |
| Richest | 0.241 (-0.126,0.608) | 0.534 (-0.163,1.232) | 0.388 (-0.114,0.890) |
| Constant | 6.079*** (5.225,6.932) | 5.852*** (4.133,7.571) | 4.469*** (3.044,5.894) |
| **Observations** | **1,082** | **327** | **405** |
| Log-likelihood | (2,072) | (627) | (709) |
| Degrees of freedom | 30 | 30 | 30 |
| P-value | - | - | - |
| Adjusted R-square | 0.167 | 0.147 | 0.270 |

* $p<0.10$,

** $p<0.05$,

*** $p<0.01$.

spending than the youngest users. User at the age of 20–30 years, 30–40 years and 40+ years spent 109.4%, 93.9% and 121.7% more OOP respectively than the youngest ones (<20 years). Unmarried users spent 99.8% more and widowed, divorced and separated users spent 138.9% less than the married users. Like the other two schemes, users in households with 4–5 persons reportedly had 34.7% less OOP than the smallest size households. The users with primary, secondary, higher secondary or higher-level education spent 91.0%, 82.8%, and 144.8% more OOP respectively than the users with no formal education. Utilisation of MTPs resulted in 111.8% more spending than the users of non MTPs and it appeared to be a strong determinant of OOP.

While comparing OOP in an adjusted model combining all schemes, overall OOP expenditure among the MHI beneficiaries were 41% higher compared to the beneficiaries of HVS. Furthermore, beneficiaries who lived in Dhaka spent 113% more OOP expenditure compared to the beneficiaries of Chattogram (S6 Table).

## Comparison of healthcare utilization and out-of-pocket payments with other schemes

In Table 8 we have compared the findings on healthcare utilisation of HVS and MHI with Urban Health Survey, 2013 and two related health insurance schemes implemented for urban and rural informal sector workers (LASP scheme) in Chandpur district and BADAS health insurance scheme of RMG workers in Dhaka. We found that any illnesses reported in HVS and MHI were close to those of LASP and BADAS health insurance scheme. In any of HVS and MHI scheme, healthcare utilisation rates were higher (HVS 96.6% and MHI 99.6%) compared to Bangladesh urban health survey (91.1%) [11]. However, LASP's healthcare utilisation was at the same level (97.7%), and RMG insurance had a lower proportion (92.1%) who sought care from any provider. Table 8 showed that healthcare seeking from MTPs were higher in

**Table 8. Comparison of illness and rate of healthcare utilization of HVS and MHI with Urban Health Survey 2013, LASP 2012, and BADAS 2015.**

| Health and healthcare | HVS | HVS | MHI | UHS[1] | LASP[2] | BADAS[3] |
|---|---|---|---|---|---|---|
| | Dhaka | Chattogram | Dhaka | | | |
| Any illness reported | 41.1 | 37.4 | 41.7 | - | 33.1 | 50.2 |
| Sought healthcare | 96.6 | 99.6 | 99.2 | 91.1 | 97.7 | 92.1 |
| Sought healthcare from MTPs | 71.6 | 91.3 | 66.0 | | 50.7 | 44.1 |
| *MNCH service* | | | | | | |
| ANC | 95 | 89 | 100 | 42.2 | - | - |
| Institutional delivery | 97.9 | 100.0 | 91.4 | 39.0 | - | - |
| PNC | 11.3 | 78 | 100 | 25.3 | - | - |
| Inpatient care | 12.1 | 4.7 | 15.1 | - | 5.4 | 2.3 |
| Sought healthcare from MTPs—by the extreme poor (poorest quintile) | 76.1 | 94.8 | 64.1 | - | 34.39 | - |
| Sought healthcare from MTPs—by the poor (2nd–5th quintile) | 72.1 | 86.8 | 67.2 | - | 49.9 | - |

[1] Urban Health Survey 2013 [11].

[2] Labour Association for Social Protection (LASP) 2012, a health insurance scheme for informal worker [27, 54].

[3] BADAS health insurance scheme for Ready-Made Garments (RMG) workers [55].

HVS and MHI schemes (71.6% in HVS, Dhaka, 91.3% in HVS, Chattogram City Corporation and 66.0% in MHI) than LASP and RMG insurance beneficiaries.

It was observed that utilisation of MTPs among the extreme poor (lowest quintile) in HVS in Dhaka and Chattogram were 76.1% and 94.8% and in MHI scheme in Dhaka was 64.1%. All these rates were higher than utilisation rate of the extreme poor enrolees of LASP scheme (34.39%). Further, the healthcare utilisation rate of the poor (2nd to 5th quintile) in HVS and MHI scheme were 72.1% (HVS Dhaka), 86.8% (HVS Chattogram) and 67.2% (MHI Dhaka), which were higher than the corresponding rate of the LASP scheme.

The proportion of pregnant mothers who received any ANC in HVS of Dhaka (95%) was higher than the urban slum population (42.2%). However, it was 89% in HVS of Chattogram City Corporation and 100% in MHI scheme. Institutional delivery rates in all three schemes under investigation were much higher than the urban slum population (39.0%). Rate of PNC utilisation was low (11.3% in HVS of Dhaka but much higher (89%) in HVS Chattogram and in MHI Dhaka (100%) compared to urban slum population (25.3%). Inpatient care utilisation rates were reportedly higher (12.1% in HVS in Dhaka and 15.1% Chattogram City Corporation) than LASP scheme (5.4%) and RMG (2.3%) insurance projects. However, it needs to be noted that inpatient care except C-section was not included in the benefits package of the schemes under investigation in this study.

OOP payments for any healthcare utilisation of the scheme enrolees in the poorest quintile were compared with the urban poorest socioeconomic group [14] and the poorest enrolees of a LASP scheme [33]. We found that HVS in Chattogram City Corporation's beneficiaries spent 1.5 Euro (per person per month) was much lesser than the urban poorest (4.3 Euro) (Fig 5). The OOP spending in HVS (1.5 Euro) and MHI (0.8 Euro) scheme in Dhaka had much lower spending compared to the urban poorest (2.6 Euro) [14]. On the other hand, all three scheme sites under investigation had much lower OOP payment than the LASP scheme (8.7 Euro).

It was thus indicated that HVS and MHI schemes secured lower OOP payment among their extreme poor enrolees.

Further, the poor segment of the enrolees in HVS in Dhaka and Chattogram and MHI scheme in Dhaka had average OOP payment of 1.7 Euro, 1.2 Euro and 2.3 Euro respectively,

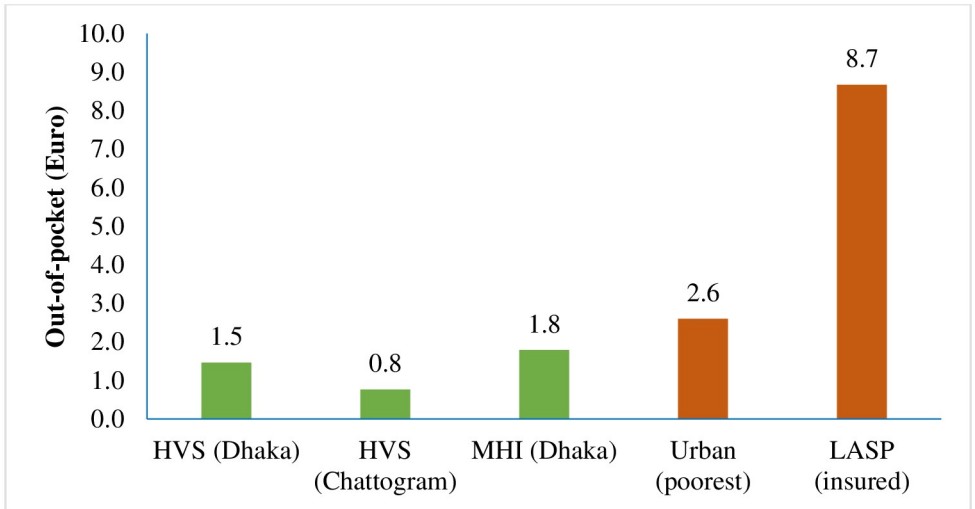

**Fig 5. Out-of-pocket spending of the extreme poor for healthcare (per person per month) in Euro.**

which were much lower than that of the nationally representative poor people (2.9 Euro) as well as of—LASP scheme (9.2 Euro) (Fig 6).

It was thus observed that HVS and MHI schemes had lower average OOP payment.

## Qualitative findings

All respondents of FGD with HVS scheme enrolees expressed their satisfaction with the services they received from the Concern- BRAC lead project facilities particularly for normal delivery, ANC and PNC. They appreciated doctor's time spent (15–20 minutes) per patient and attention of other caregivers (paramedic, midwife, health worker) towards patients.

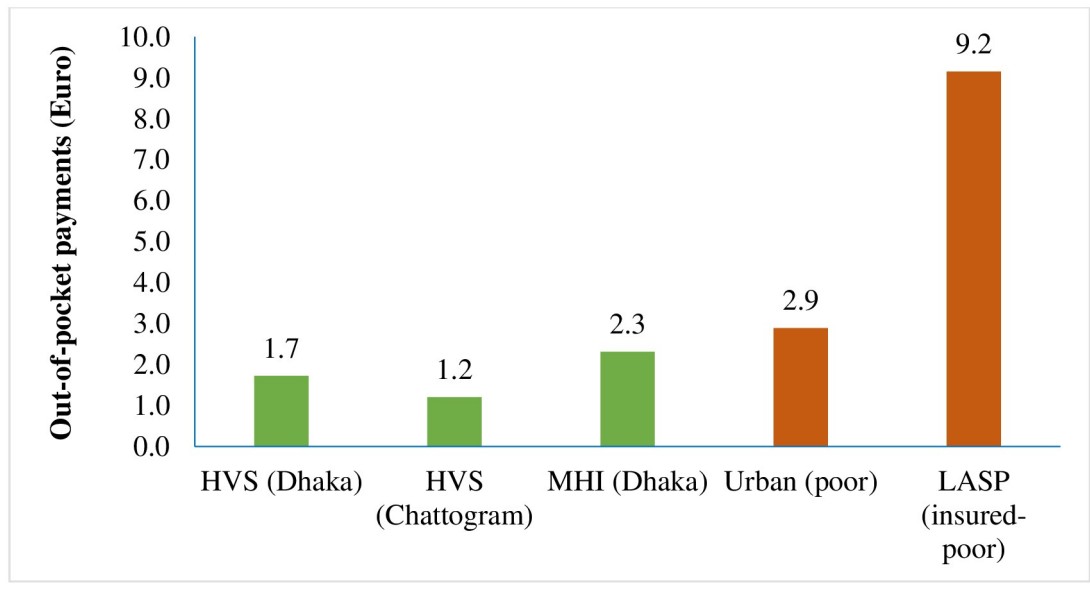

**Fig 6. Out-of-pocket spending of the poor people for healthcare (per person per month) in Euro.**

"*We are indeed happy with the services for normal delivery, ANC and PNC. Doctor's time spent per patient and other caregiver's attention towards patient is appreciable*"

- - - - - - - - -HVS card holder, Dhaka (FGD)

These respondents were informed about the benefit of the HVS card and existing system of service delivery including OPD services. However, our FGD finding indicated that their knowledge of financial limit and HVS benefits package was scarce. They mentioned the advantage of close proximity of facilities from their households, especially in Dhaka and benefit of health education messages they received from the SKs/health workers, particularly for pregnant women. However, it did not match with the PNC coverage data in case of Dhaka HVS which was low. These respondents raised concern about non-availability of all types of medicines and investigations (e.g. X-ray) at OPD, doctors for 24/7 and transport for patient referral to other hospitals. They further added that injury cases were not attended there and they had to visit other facilities for treating injury, Eye, ENT and Dental problems—often spending money from OOP.

The HVS cardholders suggested increasing manpower in the field for addressing the issue of an insufficient number of health workers. They suggested to, engage more MBBS doctors, extending OPD hours with an adequate supply of medicine. They further opined to increase

"*Increase the number of field workers, MBBS doctor, and extend OPD hours with more services and adequate supply of medicine*"

- - - - - - - -HVS card holder, Chattagram (FGD)

the benefit coverage of the card in terms of services like management of mild to moderate injury, hypertension, diabetes, Eye, ENT and dental problems.

All respondents of FGD with MHI enrolees expressed their satisfaction with the BRAC service providers for their good behaviour and supportive nature. The majority of them mentioned cleanliness of the BRAC facilities and the attendance of skilled midwives during normal delivery. Patient's satisfaction, cleanliness and availability of skilled health workers as perceived by the respondents is indicative of quality of care maintained in these facilities.

They further added that in case of referral, claim reimbursement process was quite efficient. However, there was a lack of transport for referral of serious patients to distant hospitals. These respondents expressed their concern about limited working hours of doctor at the facility- for which family members getting sick beyond doctor's duty hours had to visit other private providers.

The respondents also reported that they did not get all types of medicine from the OPD e.g., antibiotics, antihypertensive, and diabetes drugs. The MHI cardholders suggested improving the supply of quantity and types of medicine to OPD and deploy more staff including doctors. Demand for more medicines, other services (injury, hypertension, and diabetes), more workers and transport for patient referral to distant facilities were almost similar in both types of scheme enrolees.

Through regular home visits, health workers (SK) distributed leaflets and provided health, nutrition, hygiene, and family planning messages among the beneficiaries. The purpose was to improve community awareness particularly on ANC, PNC, early initiation of clostrum and exclusive breastfeeding, infant and young child feeding, immunization, and birth spacing etc.

"*We used to visit targeted households everyday to do follow ups of ANC, PNC,-if any, communicate health, nutrition and family planning messages to our cardholder households. If*

*anybody needs doctor's consultation, we bring him/her to our doctor or assist them in referral"*

- - - - -A Shaystho Kormi (SK), Dhaka (KII)

KII with Doctors revealed that they work on specific week days (3–4 days per week) and work from 9.00 am to 3 pm each day. The patients usually got medicines from *Manoshi* centres but if needed they could buy it from the nearby scheme-contracted private pharmacy and the cost was reimbursed by the scheme. Among the diagnostic tests; *Manoshi* centres had an arrangement for ultrasonogram, urine R/E and albumin, haemoglobin, blood sugar, blood grouping and cross-matching. For other tests, patients were referred to contracted medical college hospital, smiling sun and Marie Stops clinics—if required. The cost incurred was reimbursed afterwards by the scheme.

*"We are trained in ultrasonography. We can do it here using our machine. Few other routine tests are also available here. We've contracted pharmacy and referral centres around where patients can be sent for other tests or medicines, In case, we find pregnant mothers with high blood pressure or diabetes we refer them to our referral center facilitated by our staff.*

- - - - -A Medical Officer, Chattagram (KII)

Our interview with *Manoshi* centre managers revealed that the patient entry and claim reimbursement process was fully automated. They have reported on lack of understanding among HVS enrolees about the financial limitation of the card. This finding was similar to what was reported below by the ward counsellors (elected public representative).

*"There is lack of understanding among the HVS enrolees about the financial limit of the scheme they belong to. Their demand for service sometimes does not match with their card limit".*

- - - - -A Branch Manager, Chattagram (KII)

The ward counsellors in KII on HVS (Chattogram) mentioned that the enrolees were really poor having monthly income of less than 7,000 BDT.

*"The poor did not have access to free medicine previously. This programme (HVS) has changed the situation by providing financial support to the poor"*

- - - - - - -A Ward Counsellor, Chattagram (KII)

They also mentioned that there were challenges in poor identification, enrolment and updating list of target households. This happened due to undue request from some quarters and also due to high rate of migration. These respondents urged on the validation of the list of the poor from time to time as deemed necessary with the support from City Corporation. These KII respondents asked the programme officials to organize information campaign among the HVS enrolees—so that they would clearly understand the financial limit and content of the card and use it judicially.

They also requested to include all wards under this programme and suggested identifying one pharmacy or facility in each ward to serve the poor under this programme at fixed rates.

**Table 9. Distribution of provider cost and cost per beneficiary household of HVS and MHI scheme per year.**

| Inputs | HVS | | | | | | MHI | | |
|---|---|---|---|---|---|---|---|---|---|
| | Dhaka | | | Chattogram | | | Dhaka | | |
| | BDT (USD*) | Euro** | Share of total cost (%) | BDT (USD*) | Euro** | Share of total cost (%) | BDT (USD*) | Euro** | Share of total cost (%) |
| **Fixed cost** | | | | | | | | | |
| Basic training | 10562.0 (125.0) | 112.0 | 0.02 | 20067.8 (237.5) | 212.8 | 0.08 | 107838.0 (1276.2) | 1143.7 | 1.88 |
| Software (capital item) | - | - | - | - | - | - | 202097.0 (2391.7) | 2143.3 | 3.51 |
| Capital (laptop, scanner) | - | - | - | - | - | - | 8972.0 (106.2) | 95.2 | 0.16 |
| BCC (poster, leaflet) | 1584.3 (18.7) | 16.8 | 0.00 | 2218.0 (26.2) | 23.5 | 0.01 | - | - | - |
| Staff/personnel | 4412058.2 (52213.7) | 46791.8 | 6.82 | 1814691.9 (21475.6) | 19245.6 | 7.15 | 667500.6 (7899.4) | 7079.1 | 11.61 |
| **Total fixed cost** | **4424204.5 (52357.4)** | **46920.6** | **6.84** | **1836977.7 (21739.4)** | **19481.9** | **7.23** | **986407.7 (11673.5)** | **10461.3** | **17.15** |
| **Variable cost items** | | | | | | | | | |
| Incentives for marketing (SK) | 22180.2 (262.5) | 235.2 | 0.03 | 31052.3 (367.5) | 329.3 | 0.12 | 264050.0 (3124.9) | 2800.4 | 4.59 |
| Meeting | - | - | - | 84710.3 (1002.5) | 898.4 | 0.33 | 5281.0 (62.5) | 56.0 | 0.09 |
| Social mobilization (drama, folksong) | 253488.0 (2999.9) | 2688.3 | 0.39 | 506976.0 (5999.7) | 5376.7 | 2.00 | - | - | - |
| Conveyance for service provider | 76046.4 (900.0) | 806.5 | 0.12 | 237645.0 (2812.4) | 2520.3 | 0.94 | 38023.2 (450) | 403.3 | 0.66 |
| Treatment cost (consultation, medicines, tests, counseling) | 59906179.2 (708948.9) | 635330.7 | 92.62 | 22693179.4 (268558.3) | 240670.9 | 89.38 | 4456920.0 (52744.6) | 47267.5 | 77.5 |
| **Total variable cost** | **60257893.8 (713111.2)** | **639060.8** | **93.16** | **23553563 (278740.4)** | **249795.6** | **92.77** | **4764274.2 (56381.9)** | **50527.2** | **82.85** |
| **GRANT TOTAL** | **64682098.3 (765468.6)** | **685981.4** | **100.00** | **25390540.7 (300479.8)** | **269277.6** | **100.00** | **5750681.9 (68055.4)** | **60988.4** | **100.00** |
| Total number of beneficiary households | 21629 | 21629 | | 8495 | 8495 | | 4000 | 4000 | |
| **Cost per beneficiary household** | **2990.5 (35.4)** | **31.7** | | **2988.9 (35.4)** | **31.7** | | **1437.7 (17.0)** | **15.2** | |

*Exchange rate of 2019: 1 USD = 84.50 BDT;

**1 Euro = 94.29 BDT.

## Programme cost

Table 9 presents the programme costs of HVS and MHI scheme. In the Dhaka city corporation areas, the total annual cost incurred upon the provider for implementing the defined health-care for HVS was estimated to 685,981 Euro equivalents to 765,469 USD or 64,682,098 BDT for a total of 21,629 beneficiary households, which resulted in an estimated cost per beneficiary household of 32 Euro equivalent to 35 USD or 2,991 BDT in Dhaka HVS. Of the provider's total costs, 93% was attributable to treatment cost. Total fixed cost represented only 7% of the total cost of HVS in Dhaka.

For implementing HVS in 8495 beneficiary households in Chattogram, it costed a total of 269,278 Euro equivalents to 300,480 USD or 25,390,541 BDT annually on the part of the provider. The cost per household per year in Chattogram City Corporation thus appeared to 32 Euro equivalents to 35 USD or 2991 BDT. The fixed cost was 7.23% that was largely attributable to staff salary- while the variable costs constituted 92.77% of the total cost. Among the

variable cost components (Table 8), payments for treatment accounted for 89.38% of the total cost.

MHI scheme had a total cost of 60,988 Euro equivalents to 68,055 USD or 5,750,682 BDT for 4,000 beneficiary households resulting in an average cost of 15 Euro or 17 USD or 1438 BDT per beneficiary household per year. The total fixed costs constituted 17% of the total scheme costs. Within the fixed cost components, staff cost covered the largest share (12%) followed by software (4%) and training (2%). Treatment cost covered the largest share with 77.50% of the total costs.

## Discussion

Beneficiaries of both HVS and MHI schemes experienced higher healthcare utilisation and lower OOP payments compared to the corresponding population group of urban health survey of Bangladesh and other health insurance schemes. The MTP utilization was 66.0% to 91.3% in the schemes under investigation which was 50.7% and 44.1% in LASP and BADAS insurance schemes, respectively. These findings imply that reporting schemes could increase healthcare utilisation of their beneficiaries compared to the national average and related similar schemes. Further, the poor and extreme poor had higher healthcare utilisation from MTPs in HVS and MHI schemes compared to a community-based health insurance of informal workers.

The average OOP spending of the HVS in Dhaka and MHI scheme in Dhaka and HVS in Chattogram were lower than the spending reported by nationally representative urban poorest people [34]. However, in all these schemes, more services were utilised than the general urban slum population in Bangladesh, implying that the reporting schemes made progress towards UHC by securing more health care at a lower level of OOP payments. The multiple regression models showed different influential variables in different models. For instance, healthcare utilizations of MTPs were significantly explained by marital status, self-reported illness and socioeconomic status in HVS in Dhaka and sex, marital status, occupation and socioeconomic status in HVS of Chattogram City Corporation. However, the OOP spending in HVS of Dhaka and Chattogram City Corporation was significantly associated with several characteristics including age, self- reported illness, household size, educational level. It should be kept in mind in future that such factors should be considered while predicting the healthcare utilisation and expected OOP spending of schemes. It was observed that the extreme poor and poor in HVS and MHI schemes utilised more health care and paid less as OOP payments. It is indicated that the schemes have the potential to secure financial protection and increase service coverage and consequently contribute towards achieving UHC.

It should be noted that PNC was not recorded appropriately on the health card and the service recipients under HVS scheme in Dhaka could not identify the PNC visits by SK after discharge from the facility as they perceived such visits as routine ones made by SK. This might have resulted in underreporting of PNC utilisation in the survey. It should also be noted here that previous studies have found utilisation of PNC to be higher among women with a complication during pregnancy and following delivery as well as among the mothers with better economic and education levels [35]. The mothers in these small numbers of observations in the current study might have not faced such complications and they belonged to lower socioeconomic status and educational level which might have contributed to lower utilisation PNC services in some areas of HVS in Dhaka.

The qualitative investigations showed some similarities and dissimilarities of services coverage with knowledge, satisfaction and perception of beneficiaries at different sites. The beneficiaries of the scheme appeared to be happy with their access to healthcare, especially for

MNCH services, which had the advantage of the close proximity to BRAC facilities and benefited from health education messages they learned from the BRAC workers, particularly for pregnant women. However, the knowledge and perception of health education on PNC among HVS enrolees in Dhaka was not similar to the PNC coverage. The knowledge and perception of PNC seemed to be good among these beneficiaries but the coverage of PNC was low. All FGD respondents raised a similar concern about the unavailability of antihypertensive and anti-diabetic, medicines and investigations (e.g. X-ray) at OPD for all categories of patients, doctors for 24/7, more health workers and transport facilities for referral to other distant hospitals. They further added that treatment for injuries, primary care for Eye, ENT, and dental problems were not available at OPD and they had to visit other facilities for such care which incurred OOP expenditure to them. It indicated that a gap existed between the respondents' knowledge and benefits packages of the schemes- particularly of the HVS scheme. Such gap affected the provider's ability to provide care due to the financial celling of benefits package and/or unavailability of care under the package.

The time spent by healthcare providers per patient (15–20 minutes), staff behaviour, facility cleanliness and attendance of skilled midwives at the BRAC facilities were appreciated by most of the FGD respondents. This indicates that their perceived quality of care of the scheme facilities was good enough. However, this reported quality of care may not be the standard one- as it was not observed by the study team. Suggestions made by the elected ward counsellors were important to consider which include renewal of target households on a certain interval and enlarging the benefits package in future. Alongside positive remarks on the scheme activities, possible implementation challenges as mentioned by the ward counsellors need to be taken care of e.g., identifying poor households at regular intervals and engaging concerned city corporations in the identification process. This is critical because the economic conditions of these populations can improve through different interventions by the government and NGOs and local migration would highly influence the list of poor households. It is evident from responses of different KII participants that a well-organized information campaign on the schemes should be conducted among the enrolees—so that they can understand the content and financial limit of the card and use it judicially.

Health insurance is warranted in many LMICs since reliance on OOP payments for healthcare services leads to the catastrophic burden for many households. The findings from this study were similar to a number of studies that have examined the effects of health insurance/ micro health insurance schemes on healthcare utilisation and financial outcomes among members [36–41]. A number of studies have found higher utilisation of healthcare services among the insured individuals in different settings such as Bangladesh [27, 42], Congo [43], Senegal [44], India [45], and Philippines [36, 41]. However, a systematic review study found that only 14 out of 24 studies that examined the healthcare utilisation effects of health insurance observed positive outcomes [40]. Ahmed et al. (2018), in a study, found that a community-based health insurance scheme increased the utilisation of MTP among low-income informal workers in Bangladesh [27].

Hamid et al. (2011) found that micro-health insurance improves the health status of insured members which increases productivity and labour supply [46]. Jakab and Krishnan, in a review, found that 13 out of 16 studies reported insured members were likely to use more healthcare services than non-members; two studies found no difference while one found a slight decrease in healthcare use. Another study conducted by Raza et al. (2016) on community-based health insurance in India reported that the health insurance scheme had no significant effect on any utilisation outcome and there was no significant evidence of reducing financial hardship [47].

The cost per beneficiary estimate of HVS and MHI scheme was similar to the cost of MNCH services provided by *Manoshi* programme of BRAC to the slum population. Islam et al. 2010 estimated the average cost per normal delivery conducted at the selected delivery centres was 26 EURO (after inflation adjusted for the year 2019) [48]. The average cost per beneficiary was 32 Euro and 15 Euro for the HVS and MHI scheme respectively.

HVS and MHI schemes generally target extreme poor and poor people respectively and often secure specific health services, particularly MNCH in this current study, by its benefits package. As a consequence, such schemes promote equity, financial risk protection and quality of care [49]. The effectiveness of HSV and MHI schemes under investigation in this research was shown through greater healthcare benefits of the extreme poor and poor populations in the urban cities. In comparison to prevailing health benefits (utilisation of healthcare) of these target groups in general in Bangladesh (Urban Health Survey, 2013), HVS and MHI schemes increased healthcare utilisation. These benefits appeared to be higher while compared also with the LASP scheme and an BADAS scheme of similar target populations. It was further noticeable that higher utilisation of MTPs brought more people effective healthcare (evidence-based), an essential component of quality of care defined by the World Health Organization [50]. Our qualitative investigation found that the enrolees of the schemes were happy with their accessibility to healthcare through these schemes. However, an expansion of the benefits package was suggested by the enrolees in future. The OOP payments for healthcare were lower than the target population group in general in the country. The schemes thus resulted in more financial risk protection and released some disposable income of the households for spending elsewhere on goods such as food, education and clothing, which may have increased their level of welfare as non-health benefits. Further, the increase in utilisation of healthcare of the scheme enrolees contributed to equity in the society by narrowing the gaps in health benefits among socioeconomic groups (extreme poor, poor, non-poor, affluent and so forth), according to John Rawls' theory of social justice. The 'difference principle' of the theory reveals that allocating more benefits to the least advantaged person in society brings social justice [51]. The current health schemes targeted the extreme poor and poor as well as women (by MNCH), further contributing to social justice in Bangladesh. However, our methodological limitations did not allow us to quantify the exact reduction in inequity in healthcare in urban Bangladesh. The allocation of resources (costs from bringing the services to beneficiaries) appeared to be reasonable while we considered a top-down costing approach of the scheme costs. The fixed cost was quite low and the total cost per beneficiary was driven up by the variable costs, particularly treatment costs (more than 90%). The schemes, which principally used 'global funding' method (a maximum amount allocated for covering all services), should be investigated in future to find which of the provider-payment mechanisms (among capitation, result-based financing, fee for service, diagnosis-related-group and so forth) could be more appropriate for efficient and equitable use of their resources in this context [52]. The scheme possibly should consider purchasing the health services for their beneficiaries on a competitive basis from the market and the use of public health services to a greater extent for cost containment.

One of the limitations of this study is that a standard method and analysis could not be applied in this research due to practical reasons. While a standard evaluation demands a control/comparison group and baseline condition of both control and intervention groups for applying a difference-in-difference technique, this study did not have either a control group or baseline data. For this reason, we compared the scheme outcomes (like health service utilisation and OOP spending) with the indicators from the latest urban health survey of Bangladesh and two other related schemes. It should be noted here as a limitation that the voucher/or and insurance schemes (like, LASP and BADAS) often may not be fully comparable as the entitlement to benefits (different benefits packages) and co-payments (user charge while utilising the

service within benefit package are often different. Furthermore, such a comparison might have limitations since many underlying factors (confounders, like income level, housing conditions) could not be adjusted for. Another limitation is that data were collected using a recall period of six months for general illness and one year for delivery related care which might have recall bias. Furthermore, estimates on OOP expenditure were compared with other secondary sources from different time periods. However, the estimates from other sources were adjusted for inflation while comparing with the findings from this study. An earlier study found that Dhaka and Chattogram regions are more or less homogeneous based on demographic characteristics [53]. We observed difference in the household size, education level and other demographic characteristics except for age (Table 4). These differences may be influenced by different types of schemes implemented in the two regions. In HVS scheme, member need not to pay for enrolment. Whereas in MHI scheme all members need to pay except ultrapoor. Therefore, it is expected that more poor people will enrol in MHI scheme compared to HVS scheme. We have considered differences in demographic characteristics in the multiple regression model while assessing the association.

## Conclusions and recommendations

The HVS and MHI schemes appeared to be useful in increasing healthcare benefits, and led to lower OOP payments indicating the potential effectiveness of these schemes. The enrolees of the schemes were generally happy with their access to healthcare, particularly for MNCH services, while expressing their interest in having larger benefits packages in future. The schemes contributed to higher welfare to the extreme poor and poor urban people and had likely increased social justice. Costs of the services, however, appeared to be high, but there are scopes of cost containment by adapting appropriate provider-payment mechanisms in future. The schemes showed their contribution to facilitating health services to the extreme poor and poor urban populations and consequently secured financial risk protection and reduced inequity in healthcare to some extent in the society. The schemes, therefore, would contribute to achieving UHC, i.e., target 3.8 of the sustainable development goals by scaling-up in other areas of the country. Adopting a local need-based benefit package considering opinions of scheme enrolees and implementing partners and applying an appropriate mix of provider-payment mechanisms, these schemes would be useful for moving Bangladesh towards UHC.

## Supporting information

**S1 Table. Out-of-pocket payments (BDT) by scheme enrolees (in the last 6 months period) (without outliers).**
(DOCX)

**S2 Table. Overall OOP payments in Euro by scheme beneficiaries (6 months prior to survey) including outliers.**
(DOCX)

**S3 Table. Out-of-pocket payments (BDT) by scheme enrolees for delivery care (per service) (without outliers).**
(DOCX)

**S4 Table. Out-of-pocket payments in Euro by scheme enrolees for delivery care (per service) including outliers.**
(DOCX)

**S5 Table. Factors associated with the utilization of medically trained providers.**
(DOCX)

**S6 Table. Factors associated with the log of out-of-pocket healthcare expenditure.**
(DOCX)

**S1 File. Focus group discussion and key informant interview guideline.**
(PDF)

**S2 File. Dataset.**
(DTA)

## Acknowledgments

The authors remain thankful to all partners of the programme, data collectors and supervisors for their hard work and to the survey respondents and other study participants for their valuable time.

## Author Contributions

**Conceptualization:** Sayem Ahmed, Jahangir A. M. Khan, Ziaul Islam.

**Data curation:** Sayem Ahmed, Nausad Ali, Mohammad Wahid Ahmed.

**Formal analysis:** Sayem Ahmed, Md. Zahid Hasan, Nausad Ali, Mohammad Wahid Ahmed, Jahangir A. M. Khan, Ziaul Islam.

**Funding acquisition:** Jahangir A. M. Khan, Ziaul Islam.

**Investigation:** Sayem Ahmed, Jahangir A. M. Khan, Ziaul Islam.

**Methodology:** Sayem Ahmed, Nausad Ali, Jahangir A. M. Khan, Ziaul Islam.

**Project administration:** Sayem Ahmed, Ziaul Islam.

**Resources:** Christine Bousquet.

**Supervision:** Sayem Ahmed, Nausad Ali, Mohammad Wahid Ahmed, Sadia Shabnam, Morseda Chowdhury, Jahangir A. M. Khan, Ziaul Islam.

**Validation:** Nausad Ali, Emranul Haq, Morseda Chowdhury.

**Visualization:** Emranul Haq, Sadia Shabnam, Morseda Chowdhury, Breda Gahan, Christine Bousquet, Jahangir A. M. Khan, Ziaul Islam.

**Writing – original draft:** Sayem Ahmed, Md. Zahid Hasan, Nausad Ali, Mohammad Wahid Ahmed, Jahangir A. M. Khan, Ziaul Islam.

**Writing – review & editing:** Sayem Ahmed, Md. Zahid Hasan, Nausad Ali, Mohammad Wahid Ahmed, Emranul Haq, Sadia Shabnam, Morseda Chowdhury, Breda Gahan, Christine Bousquet, Jahangir A. M. Khan, Ziaul Islam.

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
