## [Decision Letter · Decision Letter 0]

1 Jan 2021

PONE-D-20-22715

Potential Effectiveness of Health Voucher Scheme and Micro-Health Insurance Scheme to Support the Poor and Extreme Poor in Selected Urban Areas of Bangladesh – An Assessment using a mixed-method approach

PLOS ONE

Dear Dr. Ahmed,

Thank you for submitting your manuscript to PLOS ONE. After careful consideration, we feel that it has merit but does not fully meet PLOS ONE’s publication criteria as it currently stands. Therefore, we invite you to submit a revised version of the manuscript that addresses the points raised during the review process.

Please review the comments from the reviewers including the comments from me as listed below. At the minimum, the potential errors in reporting background information (as indicated by the reviewers) and editorial changes should be completed. Methodology section should be rewritten to provide additional information needed. Re-running the empirical models is not absolutely essential but will be useful in improving the quality of the paper.

We look forward to receiving your revised manuscript.

Kind regards,

M. Mahmud Khan

Academic Editor

PLOS ONE

Journal Requirements:

2. Please specify whether an interview guide was used to interview the participants in your study.

If yes, please describe and/or include a copy as a Supporting Information file.

3. Thank you for including your ethics statement:  "The study was approved by the institutional review board of icddr,b under the protocol number PR-19084. Informed written consent was obtained from the respondents before conducting interview. ".   

'I have read the journal's policy and the authors of this manuscript have the following competing interests:

Sayem Ahmed – none declared

Md. Zahid Hasan – none declared

Nausad Ali – none declared

Mohammad Wahid Ahmed – none declared

Emranul Haq – works at the Concern Worldwide Bangladesh

Sadia Shabnam – works at the scheme implementing organization, BRAC

Morseda Chowdhury – works at the scheme implementing organization, BRAC

Breda Gahan – works at the funding organization, Concern Worldwide

Christine Bousquet – works at the funding organization, Concern Worldwide

Jahangir A. M. Khan – none declared

Ziaul Islam – none declared'

a. Please confirm that this does not alter your adherence to all PLOS ONE policies on sharing data and materials, by including the following statement: "This does not alter our adherence to  PLOS ONE policies on sharing data and materials.” (as detailed online in our guide for authors http://journals.plos.org/plosone/s/competing-interests).  If there are restrictions on sharing of data and/or materials, please state these.

Please note that we cannot proceed with consideration of your article until this information has been declared.

6. Please ensure that you refer to Figure 7 in your text as, if accepted, production will need this reference to link the reader to the figure.

Additional Editor Comments:

Thanks for submitting the paper for possible publication in PLOS One. We have now received two reviews and I agree with the first reviewer that the paper needs significant revisions. The context of Bangladesh and slum population as percent of total urban population needs correction as reviewer 1 has mentioned. Using 2010 survey on health expenditure for comparative purposes will be highly biased due to the time lag between the national survey and the data collected for this study. As suggested, please use 2016 survey for comparative purposes. In the description of the survey, authors should clearly indicate the survey time frame (when did the survey actually carried out). Empirical models estimated should be mentioned so that readers can interpret/understand the importance of the parameters estimated. In addition to the specific comments from the reviewers, I have few additional comments:

1. The dependent variables appear to be common in all the three data sets (HVS Dhaka, HVS Chittagong and MHI Dhaka). Since one of the important objectives of the study is to compare the outcomes across these areas and programs, no empirical model has been estimated to test the differences. It is not clear why the quantitative data from all the two programs and two areas cannot be combined in a grand model and then using area and program dummies as independent variables to test the differences directly rather than comparing the averages in an ad-hoc manner.

2. From the information provided in the tables, it appears that the surveyed households/individuals in Chittagong are quite different from surveyed units in Dhaka. It is important to mention possible reasons for the differences.

3. There appears to be a problem in the distribution of individuals by asset quintiles as presented in Table 4. I thought that the percent in each quintile has been defined separately by HVS Dhaka, HVS Chittagong and MHI Dhaka so that sum of column percentages will be 100%. That appears to be the case for HVS Dhaka and MHI Dhaka but not for HVS Chittagong. Please explain the reason for this discrepancy.

Reviewers' comments:

Reviewer's Responses to Questions

**Comments to the Author**

1. Is the manuscript technically sound, and do the data support the conclusions?

Reviewer #1: Yes

Reviewer #2: Yes

2. Has the statistical analysis been performed appropriately and rigorously? 

Reviewer #1: Yes

Reviewer #2: Yes

3. Have the authors made all data underlying the findings in their manuscript fully available?

Reviewer #1: No

Reviewer #2: Yes

4. Is the manuscript presented in an intelligible fashion and written in standard English?

Reviewer #1: Yes

Reviewer #2: Yes

5. Review Comments to the Author

Reviewer #1: The manuscript provides valuable insights for understanding the roles of health voucher and micro-health insurance schemes to improve MNCH services from skilled providers in urban areas. The manuscript is original and reads well, and the findings are important to public health researchers, academics, and policymakers. As the PLoS series' flagship journal, PLoS One is the appropriate place to publish this paper because it can reach each target group.

I would recommend publication of this manuscript in the journal after some major revisions as specified below:

1. p.4, lines 76-77: Please update this statement. The World Bank source (https://data.worldbank.org/indicator/EN.POP.SLUM.UR.ZS?locations=BD) states that 47.2% of the urban population in 2018 lived in slums (original data source: UN-HABITAT). Also, BBS estimated 2,232,114 slum dwellers in the 2014 slum census, which was 6.33% of the country's urban population. (source: Census of Slum Areas and Floating Population 2014: Final Report, p.31)

2. p.6, lines 141-143: Please provide a reference for the statement “Prior to these …. and large population size.”

3. p.8, lines 191-192: Authors stated that “OOP spending information 191 was compared with the Bangladesh Household Income Expenditure Survey 2010.” The final report of the latest (2016) round of HIES was published in June 2019, and its data is available from BBS. I'd recommend the authors to use the 2016 HIES data for comparison, which would be more appropriate.

4. p.10, lines 273-275: The authors should consider providing a rationale for deleting outliers. Unless these observations are not the result of a measurement or some other form of errors, the data without the outliers are not representative of the study sample, and therefore all inference will be more or less meaningless.

5. pp.10-11, lines 276-280: The analyses used two different regression models but did not have any information on these except stating “multiple regression analyses were conducted.” The authors need to provide regression model specifications and description of dependent and exploratory variables in this sub-section (i.e., data analysis).

6. p.11, lines 309-315: Similar texts/statements repeated in lines 735-746. Please consider a) removing the texts from p.11, and b) having a section on study limitations at the end of the discussion and note this limitation there (along with other limitations, viz., potential recall bias, comparison of estimates with other secondary sources from the different time period, etc.).

7. In-text citation and the references section require thorough review and correction. For, example, lines 256, 685, 688, 705, 781, 783, 796, etc.

Minor comments:

1. Bangladesh Diabetic Somity should be replaced by their official name Diabetic Association of Bangladesh (BADAS)

2. Consider deleting the secondary data sources used (UHS 2013 and HIES 2010) from the list of keywords

3. Check and correct the use of abbreviations throughout the manuscript (i.e., full elaboration at the first use, and abbreviations thereafter). For example, lines 182, 190, Table 1, Table 2, etc.

4. p.8, line 198: Table 1 should be replaced by Table 2

5. p.9, lines 221-223: Please consider stating the affiliations of the doctors and managers interviewed as KII (from BRAC, private clinics, etc.)

6. Check and correct the use of punctuation throughout the manuscript. For example, redundant use of comma (,) in line 299, missing full-stop (.) in line 318, missing comma (,) in line 617, redundant (.) in line 667, etc.

Reviewer #2: I have a few comments/questions that I think should be addressed properly.

Comments:

Abstract:

1. Results: What are the components of MNCH services? Please include the services under MNCH.

2. Conclusion: Remove “;” and put “,” in line number 63 (page 3).

3. Can you please elaborate this statement “….., but there are potential of cost containment by investigating and adapting appropriate provider payment mechanism”.

Introduction:

1. Please take care of the punctuations in the first paragraph (missing coma in several statements).

2. “82.4% of slum dwellers received health care from informal providers and such care significantly resulted in adverse effects on health”- why is that? Please describe.

3. Line 104 “health services leads to …..” should be “health services lead to……”. Please go through the article thoroughly and check for grammatical discrepancies.

4. You provided acronym in line number 105 however, did not use it properly. Please ensure consistency.

5. Provide reference for line number 122 (target 3.8 of the Sustainable Development Goals of the United Nations).

6. “The strategy of Bangladesh suggested that the financing of healthcare of extreme poor should be managed by government’s tax revenue and of the low-income informal sector workers by community-based health insurance.” Any reference?

Materials and Methods:

1. Monetary values not “monitory values” (line 192).

2. Check for table numbering.

3. Apart from cardholders number variations in different locations did you consider any other factors for sample size calculations (PPS) for different locations e.g., population density, NGOs, public or private healthcare facilities. If not, why?

4. Did you pretest the semi-structured questionnaire before the actual data collection with the trained data collectors?

5. Did cross-sectional survey and qualitative data collection done by the same data collectors?

6. Why did you conduct 2 FGDs in Rasulbagh and Islampur and excluded the other 2 areas for MHI scheme?

7. Was interview guidelines prepared separately for FGDs and KIIs? Did you pilot tested these guidelines and revised accordingly based on the findings? If not, why?

8. Qualitative data collection should be elaborated more (e,g., who conducted the main interviews, any note-takers, recording proceedings).

9. Please mention all the items that were included in ‘fixed’ and ‘variable’ costs.

10. Did you include transport cost (for healthcare seeking purpose) of the beneficiaries during the costing analysis? If not, why? As the population per argument was poor/extremely poor, this is supposed to be an important cost to consider.

11. Transcripts were done verbatim? Did anyone check for consistencies and errors?

Results and discussion:

1. Line 338-341 is not grammatically correct. Need to revise.

2. Please check the article thoroughly for caps lock (Normal delivery, line 361), grammatical issues and proper sentence making.

3. Line 385-386 is not clear. Need revision.

4. Line 393 add ‘and’ after 16.2 Euro.

5. Line 394-396 is not clear. Please revise.

6. Please revisit table 7 (Factors of out-of-pocket (logged) payments by health schemes) as well as the ‘Determinants of out-of-pocket payments’ description. Look out for the interpretations and significant association and detail out clearly for each of the schemes and study sites (Page 23-23 and Page 16).

7. What is (in %) of Table 8 title? Items of Table 8 (superscript) not properly done. Missing values for UHS. You need to check the title names and make proper title of the tables (including the survey names, year etc).

Qualitative findings:

1. First quotes should be properly placed after the relevant paragraph. It was misplaced.

2. Participants ID’s were not clearly stated (i.e., from Dhaka or Chattogram). Kindly include district names for all quotes.

3. Line 541 - Please revise the sentence.

4. Line 546-550 – Rephrase and correct the spelling.

5. Please look for grammatical issues (e.g., spelling, uppercase/lowercase issue, punctuations) throughout the document.

6. Line 594 – ‘32 Euro equivalent to 35 USD or 2,991 BDT in Dhaka’, however in line 600 it is ‘32 Euro equivalents to 35 USD or 2989 BDT’. Why it is different?

Discussion:

1. Line 647-648 - What do you mean by ‘similarities 647 and dissimilarities of demands raised by the enrolees and a mismatch of PNC coverage’. Can you explain?

2. Your statement in line 652-653 ‘their knowledge and perception of health education on PNC did not match with the PNC coverage among HVS enrolees in Dhaka where it appeared to be low.’ Is confusing. Please rephrase.

3. Please revise the statement from line 657-662.

4. What do you mean by ‘as they relate to the renewal of target households and enlarging the service package in future’ in line 668-669.

5. Apart from the standard method and analysis limitation what are the other limitations in this study? What are the strengths of this study?

Acknowledgment:

1. What is ‘Governments of Bangladesh’ (line 771)?

6. PLOS authors have the option to publish the peer review history of their article (what does this mean?). If published, this will include your full peer review and any attached files.

Reviewer #1: **Yes: **Karar Zunaid Ahsan

Reviewer #2: No

---

## [Author Response · Author response to Decision Letter 0]

23 Jun 2021

Reviewers' comments:

Reviewer #1: 

The manuscript provides valuable insights for understanding the roles of health voucher and micro-health insurance schemes to improve MNCH services from skilled providers in urban areas. The manuscript is original and reads well, and the findings are important to public health researchers, academics, and policymakers. As the PLoS series' flagship journal, PLoS One is the appropriate place to publish this paper because it can reach each target group.

I would recommend publication of this manuscript in the journal after some major revisions as specified below:

Comment 1. p.4, lines 76-77: Please update this statement. The World Bank source (https://data.worldbank.org/indicator/EN.POP.SLUM.UR.ZS?locations=BD) states that 47.2% of the urban population in 2018 lived in slums (original data source: UN-HABITAT). Also, BBS estimated 2,232,114 slum dwellers in the 2014 slum census, which was 6.33% of the country's urban population. (source: Census of Slum Areas and Floating Population 2014: Final Report, p.31)

Response: We updated the statistics using The World Bank source as suggested by the reviewer-“In Bangladesh 47.2% of the total urban population live in slums according to The World Bank”. (Please see page 4, first para).

Comment 2. p.6, lines 141-143: Please provide a reference for the statement “Prior to these …. and large population size.”

Response: We cited following report published by Concern Worldwide for this,

Concern Worldwide. Lessons From The City : Concern ’ S Work In Urban Areas. Dublin, 2016.

Comment 3. p.8, lines 191-192: Authors stated that “OOP spending information 191 was compared with the Bangladesh Household Income Expenditure Survey 2010.” The final report of the latest (2016) round of HIES was published in June 2019, and its data is available from BBS. I'd recommend the authors to use the 2016 HIES data for comparison, which would be more appropriate.

Response: We thank the reviewer for identifying this. However, we removed the comparison with the OOP estimates of the current study with household income and expenditure survey 2010 estimates later. Therefore we do not need to use HIES 2016 data for comparison. We mentioned of household income and expenditure survey in the text mistakenly. We have now removed the lines. (Page 8, first para)

Comment 4. p.10, lines 273-275: The authors should consider providing a rationale for deleting outliers. Unless these observations are not the result of a measurement or some other form of errors, the data without the outliers are not representative of the study sample, and therefore all inference will be more or less meaningless.

Response: Due to outliers the average OOP payments estimate was affected. Therefore, we have removed outliers following statistical procedure proposed by Hadi 1994. We have now presented estimates of OOP payments both with and without outliers so that reader can get estimate which is representative of the study sample (Please see supplementary table S1-table S2.1).

Comment 5. pp.10-11, lines 276-280: The analyses used two different regression models but did not have any information on these except stating “multiple regression analyses were conducted.” The authors need to provide regression model specifications and description of dependent and exploratory variables in this sub-section (i.e., data analysis).

Response: We have used two types of multiple regression models 1. Log-linear and 2. Binary logistic. We have added following text on the model specification in the manuscript (Please see page 11, last para). 

In the first model, we used a natural log form of the OOP expenditure ln (Y1) as dependent variable. The model was specified as follows;

ln 〖(Y〗_1i)=β_0+β_1 X_1i+β_2 X_2i+β_3 X_3i+⋯+ε_i………..(1)

where β0 is a constant, X1 X2, X3, . . . denote control variables, β1, β2, β3, . . . represent the estimated coefficients, and εi is the random error term of the model. In the second model, we used a binary dependent variable and the model was specified as follows

Logit 〖(Y〗_ki)=θ_0+θ_1 X_1i+θ_2 X_2i+θ_3 X_3i+⋯+u_i………..(2)

Where, Yk is the dependent variable for utilizing healthcare from medically trained provider (MTP) and coded as binary (0= No 1=Yes). θ0 is a constant, X1 X2, X3, . . . denote control variables, θ1 θ2, θ3 . . . represent the estimated coefficients, and ui is the random error term of the model. 

Comment 6. p.11, lines 309-315: Similar texts/statements repeated in lines 735-746. Please consider a) removing the texts from p.11, and b) having a section on study limitations at the end of the discussion and note this limitation there (along with other limitations, viz., potential recall bias, comparison of estimates with other secondary sources from the different time period, etc.).

Response: We removed the study limitation from method section (page 13). We have now added all limitations of the study at the end of discussion section (Please see page 29, last para and page 30 first para). Following text are added on recall bias as suggested by the reviewer, 

 “Another limitation is that data were collected with a recall period of six months for general illness and one year for delivery related care which might have create recall bias. Furthermore, estimates on OOP expenditure were compared with other secondary sources from different time periods. However, the estimates from other sources were adjusted for inflation while comparing with the findings from this study” (Please see page 30 first para).

Comment 7. In-text citation and the references section require thorough review and correction. For, example, lines 256, 685, 688, 705, 781, 783, 796.

Response: Many thanks for identified this important issue. We have now updated the references and citations in the manuscript. 

Minor comments:

Comment 8. Bangladesh Diabetic Somity should be replaced by their official name Diabetic Association of Bangladesh (BADAS)

Response: We have now replaced the text as per suggestion. (page 7; 2nd para)

Comment 9. Consider deleting the secondary data sources used (UHS 2013 and HIES 2010) from the list of keywords

Response: We have now removed the data sources as per suggestion. (page 3)

Comment 10. Check and correct the use of abbreviations throughout the manuscript (i.e., full elaboration at the first use, and abbreviations thereafter). For example, lines 182, 190, Table 1, Table 2, etc. 

Response: We have now updated the abbreviations throughout the manuscript. 

Comment 11. p.8, line 198: Table 1 should be replaced by Table 2.

Response: We have now updated this (page 8. 2nd para).

Comment 12. p.9, lines 221-223: Please consider stating the affiliations of the doctors and managers interviewed as KII (from BRAC, private clinics, etc.)

Response: We have interviewed two Shaystho Kormis (health workers), two Branch Managers, and two Programme Managers from BRAC and two ward counsellors of Chattogram City-corporation. (Page 9, para 2)

Comment 13. Check and correct the use of punctuation throughout the manuscript. For example, redundant use of comma (,) in line 299, missing full-stop (.) in line 318, missing comma (,) in line 617, redundant (.) in line 667, etc.

Response: We have now updated incorrect punctuation throughout the manuscript.

 

Reviewer #2: 

I have a few comments/questions that I think should be addressed properly.

Abstract:

Comment 1. Results: What are the components of MNCH services? Please include the services under MNCH.

Response: The MNCH include maternal, neonatal, and child healthcare services. We have now elaborated this now in the abstract. 

Comment 2. Conclusion: Remove “;” and put “,” in line number 63 (page 3).

Response: We have now updated this as per suggestion. (page 3)

Comment 3. Can you please elaborate this statement “….., but there are potential of cost containment by investigating and adapting appropriate provider payment mechanism”.

Response: We updated the statement as below,

The provider’s costs of the schemes were reasonable; however,, but there are potential of cost containment by investigating and purchasing the health services for their beneficiaries in a competitive basis from the market. (page 3, first para)

Introduction:

Comment 4. Please take care of the punctuations in the first paragraph (missing coma in several statements).

Response: We have corrected punctuations in the first paragraph of the manuscript. (page 4, first para)

Comment 5. “82.4% of slum dwellers received health care from informal providers and such care significantly resulted in adverse effects on health”- why is that? Please describe.

Response: Healthcare utilization from informal providers can have adverse effect on health because of non-guideline-based treatment and overuse of drugs. 

We have now revised the lines as follows “Evidences showed that 82.4% of slum dwellers received health care from informal providers (Jahan et al., 2015; NIPORT, 2013). Healthcare utilization from informal providers can have adverse effect on health because of non-guideline-based treatment and overuse of drugs (Caldwell et al., 2002, 2014)” (Please see page 4, 2nd para) 

Comment 6. Line 104 “health services leads to …..” should be “health services lead to……”. Please go through the article thoroughly and check for grammatical discrepancies.

Response: We have corrected this statement as suggested (page 5, first para). The manuscript is now thoroughly checked for grammatical discrepancies.

Comment 7. You provided acronym in line number 105 however, did not use it properly. Please ensure consistency.

Response: We have now used the acronym “OOP consistently in the manuscript.

Comment 8. Provide reference for line number 122 (target 3.8 of the Sustainable Development Goals of the United Nations).

Response: We have added following reference for this (Page 5, last para), 

United Nations. (2018). Transforming Our World: The 2030 Agenda for Sustainable Development. In http://www.un.org/millenniumgoals/ [cited 2015 July 30. Springer Publishing Company. https://doi.org/10.1891/9780826190123.ap02

Comment 9. “The strategy of Bangladesh suggested that the financing of healthcare of extreme poor should be managed by government’s tax revenue and of the low-income informal sector workers by community-based health insurance.” Any reference?

Response: We have added the following reference now (Please see page 7),

MoHFW. (2012). Expanding Social Protection for Health : Towards Universal Coverage Health Care. 

Materials and Methods:

Comment 10. Monetary values not “monitory values” (line 192).

Response: Many thanks. Now we have updated this. (Page 8, first para)

Comment 11. Check for table numbering.

Response: We have now corrected the table numbering (page 8).

Comment 12. Apart from cardholders number variations in different locations did you consider any other factors for sample size calculations (PPS) for different locations e.g., population density, NGOs, public or private healthcare facilities. If not, why?

Response: We considered cardholders number variations in different locations did you consider any other factors for sample size calculations (PPS) for different locations. This is because the cardholders were the respondent of this study. We did not selected the sample from the whole population. Types of healthcare facilities utilized by the individual were adjusted in the analysis not during the sample design. (page 8, last para)

Comment 13. Did you pretest the semi-structured questionnaire before the actual data collection with the trained data collectors?

Response: Yes, we have trained the data collectors and pretested the semi structured questionnaire before the start of final data collection. The pretesting was done in a slum of Dhaka City. We have added the following lines “The questionnaire, guidelines, and checklists were pretested before starting of the final data collection. Data was collected during August to September 2019.” (Please see page-10, first para)

Comment 14. Did cross-sectional survey and qualitative data collection done by the same data collectors?

Response: No, the qualitative data and survey data were collected by separate data collectors. Qualitative data were collected by the investigators of the project. 

Comment 15. Why did you conduct 2 FGDs in Rasulbagh and Islampur and excluded the other 2 areas for MHI scheme?

Response: Two FGDs were conducted in two branch office of BRAC. In there, Rasulbag branch office covered two areas: Rasulbag and Rayer Bazar and Islampur Branch covered Islampur and Kamrangir Char areas. However, the participants of FGDs were come from the two areas from each branch office. We have chosen one branch office per site since there are not much difference between the branches in terms of program implementation and services. 

Comment 16. Was interview guidelines prepared separately for FGDs and KIIs? Did you pilot tested these guidelines and revised accordingly based on the findings? If not, why?

Response: Yes, we developed separate guidelines for FGDs and KIIs. We have piloted the FGDs and KIIs guidelines before the final interview. We have added the following lines in the manuscript ““The questionnaire, guidelines, and checklists were pretested before starting of the final data collection. Data was collected during August to September 2019.” (Please see Page-10, first para) 

Comment 17. Qualitative data collection should be elaborated more (e,g., who conducted the main interviews, any note-takers, recording proceedings).

Response: We have now elaborated qualitative data collection as below, 

“The FGDs and KIIs were conducted by the study investigators and a note taker took notes of the interviews. d All the FGDs and KIIs sessions were audio recorded with prior permission of the participants.” Please see page 9, 3rd para 

Comment 18. Please mention all the items that were included in ‘fixed’ and ‘variable’ costs.

Response: We have defined the fixed costs and variable costs along with the items in the manuscript as “The costs that didn’t vary with the number of beneficiaries were considered as fixed costs e.g. staff training, capital items (laptops, scanner etc.), software, staff salary, BCC materials (poster, leaflets). Costs that varied with the number of beneficiaries were considered as variable costs (e.g., incentive of SKs, reimbursement of claims/treatment cost, diagnostic tests, meetings, and periodic social mobilization activities).” (Please see page 10, 2nd para)

Comment 19. Did you include transport cost (for healthcare seeking purpose) of the beneficiaries during the costing analysis? If not, why? As the population per argument was poor/extremely poor, this is supposed to be an important cost to consider.

Response: While estimating the OOP payments we have included the transportation costs as a cost component. Please see figure 2.

Comment 20. Transcripts were done verbatim? Did anyone check for consistencies and errors?

Response: Yes, the verbatim transcriptions were checked by investigators for consistency. We have added a line “The transcriptions were then checked by other investigators for consistencies and errors.” (Please see page 13, first para) 

Results and discussion:

Comment 21. Line 338-341 is not grammatically correct. Need to revise.

Response: We have revised the lines as follows “Fifty eight percent of the households had 4-5 members in all three schemes. Very few beneficiaries had disability and were members of other NGO/cooperatives. Majority of the beneficiaries had secondary level schooling (9-10 years) with a share of 35.3% in Dhaka HVS scheme, 35.9% in Chattogram city-corporation HVS scheme and 30.7% in Dhaka MHI scheme.” Please see page 14, 2nd para

Comment 22. Please check the article thoroughly for caps lock (Normal delivery, line 361), grammatical issues and proper sentence making.

Response: We have checked the article and updated inconsistency in capital alphabets and grammatical issues where necessary. 

Comment 23. Line 385-386 is not clear. Need revision.

Response: Many thanks for this suggestion. We have now revised the lines as follows “Variation in OOP payments was observed across the socioeconomic groups in all the three studied schemes. However, in HVS of Chattogram city-corporation and MHI of Dhaka, OOP payments gradually increased with the increase of the socioeconomic status of the beneficiaries.” Please see page 16, first para

Comment 24. Line 393 add ‘and’ after 16.2 Euro.

Response: We have now updated this.

Comment 25. Line 394-396 is not clear. Please revise.

Response: We have now revised the lines as follows “In Chattogram city-corporation, the average OOP spending by enrolees of HVS for ANC and PNC was similar (1.1 Euro, 1.8 USD, 104 BDT). However, these beneficiaries incurred 8.9 Euro (10 USD, 839 BDT) for normal delivery and 47.5 Euro (53.3 USD, 4479 BDT) for C-section”( Please see page 16, 2nd para). 

Comment 26. Please revisit table 7 (Factors of out-of-pocket (logged) payments by health schemes) as well as the ‘Determinants of out-of-pocket payments’ description. Look out for the interpretations and significant association and detail out clearly for each of the schemes and study sites (Page 23-23 and Page 16).

Response: We have now revised the interpretation of Table 7.(Please see page 17 last para, page 18 first para) 

Comment 27. What is (in %) of Table 8 title? Items of Table 8 (superscript) not properly done. Missing values for UHS. You need to check the title names and make proper title of the tables (including the survey names, year etc).

Response: We have now revised the title as “Comparison of illness and healthcare utilization rate of HVS and MHI with urban health survey 2013, LASP 2012, and BADAS 2015.” Please see title of table 8.

Qualitative findings:

Comment 28. First quotes should be properly placed after the relevant paragraph. It was misplaced.

Response: We have placed the first quote accordingly. Please see page 21 

Comment 29. Participants ID’s were not clearly stated (i.e., from Dhaka or Chattogram). Kindly include district names for all quotes.

Response: We have included district names (Dhaka or Chattogram) in this version. (page 21-24)

Comment 30. Line 541 - Please revise the sentence.

Response: We have now revised the line as “The respondents also reported that they did not get all types of medicine from the OPD e.g., antibiotics, antihypertensive and antidiabetic drugs” Please see page 22, para 5

Comment 31. Line 546-550 – Rephrase and correct the spelling.

Response: We have now revised the lines as follows “Through regular home visits, health workers (SK) distributed leaflets and provided health, nutrition, hygiene, and family planning messages among the beneficiaries. The purpose was to improve community awareness particularly on ANC, PNC, early initiation of breastfeeding and exclusive breastfeeding, infant and young child feeding, immunization, and birth spacing etc.” (Please see page 22, last para)

Comment 32. Please look for grammatical issues (e.g., spelling, uppercase/lowercase issue, punctuations) throughout the document.

Response: We have now checked the spelling, uppercase/lowercase issues throughout the document and updated.

Comment 33. Line 594 – ‘32 Euro equivalent to 35 USD or 2,991 BDT in Dhaka’, however in line 600 it is ‘32 Euro equivalents to 35 USD or 2989 BDT’. Why it is different?

Response: We have corrected this to 2991 in this version. (Page 24, last para)

Discussion:

Comment 34. Line 647-648 - What do you mean by ‘similarities 647 and dissimilarities of demands raised by the enrolees and a mismatch of PNC coverage’. Can you explain?

Response: We have now revised the line. We intended to mention the similarities and dissimilarities of service coverage with the knowledge, satisfaction, and perception of the beneficiaries. For example, the PNC coverage was low in Dhaka HVS but the knowledge and perception were not low among the beneficiaries. We have now revised the line as follows. “The qualitative investigations showed some similarities and dissimilarities of service coverage with knowledge, satisfaction and perception of beneficiaries at different sites.” Please see page 26, 3rd para 

Comment 35. Your statement in line 652-653 ‘their knowledge and perception of health education on PNC did not match with the PNC coverage among HVS enrolees in Dhaka where it appeared to be low.’ Is confusing. Please rephrase.

Response: We have now revised the line as follows “However, the knowledge and perception of health education on PNC among HVS enrolees in Dhaka was not similar to the PNC coverage. The knowledge and perception of PNC seemed to be good among these beneficiaries but the coverage of PNC was low.” Please see page 26, last para

Comment 36. Please revise the statement from line 657-662.

Response: We have now revised the lines as per suggestion as follows “They further added that treatment for injuries, primary care for Eye, ENT, and dental problems were not available at OPD and they had to visit other facilities for such care which incurred OOP expenditure to them. It indicated that a gap existed between respondents’ knowledge and benefits packages of the schemes exists; particularly of the HVS scheme. Such gap affected the provider’s ability to provide care due to the financial celling of benefits package and/or unavailability of care under the package.” Please see page 26, last para

Comment 37. What do you mean by ‘as they relate to the renewal of target households and enlarging the service package in future’ in line 668-669.

Response: The ward counsellors emphasized on renewing the lists of target households as many the existing households might have improved socioeconomic status overtime. Furthermore, the current benefit package could be improved to reduce the OOP expenditure for unoffered primary cares. We have revised the line as follows “Suggestions made by the elected ward counsellors were important to consider which include renewal of target households on a certain interval and enlarging the benefit package in future.” Please see page 27, second para

Comment 38. Apart from the standard method and analysis limitation what are the other limitations in this study? What are the strengths of this study?

Response: We have now added a limitation and strength of the study in the last para of discussion section. Please see page 29, last para 

Acknowledgment:

Comment 39. What is ‘Governments of Bangladesh’ (line 771)?

Response: We corrected this as “Government of Bangladesh” in this version. Please see page 31, first para

Additional Editor Comments:

Comment 1. Thanks for submitting the paper for possible publication in PLOS One. We have now received two reviews and I agree with the first reviewer that the paper needs significant revisions. The context of Bangladesh and slum population as percent of total urban population needs correction as reviewer 1 has mentioned. 

Using 2010 survey on health expenditure for comparative purposes will be highly biased due to the time lag between the national survey and the data collected for this study. As suggested, please use 2016 survey for comparative purposes. 

In the description of the survey, authors should clearly indicate the survey time frame (when did the survey carried out). 

Empirical models estimated should be mentioned so that readers can interpret/understand the importance of the parameters estimated. 

Response: We thank the editor for providing us an opportunity to revise the manuscript. We have now updated the statistics using The World Bank source as suggested by the reviewer as “In Bangladesh 47.2% of the total urban population live in slums according to The World Bank”. (Please see page 4, first para). 

We appreciate the suggestion from the reviewer regarding the use of the latest household income and expenditure survey data for comparing the study findings. However, we didn’t’ compare the OOP estimates of the current study with household income and expenditure survey estimates 2010 rather we compare this with the urban health survey 2013 and similar two others scheme operated in the urban area. We have now removed the household income expenditure survey 2010 from the manuscript.

Data were collected from August to September 2019. We have now added the data collection period. (Please see page 10, first para)

We have used two types of multiple regression models 1. Log-linear and 2. Binary logistic. We have now added the model specification in the manuscript (Please see page 11). 

In addition to the specific comments from the reviewers, I have few additional comments:

Comment 2. The dependent variables appear to be common in all the three data sets (HVS Dhaka, HVS Chittagong and MHI Dhaka). Since one of the important objectives of the study is to compare the outcomes across these areas and programs, no empirical model has been estimated to test the differences. It is not clear why the quantitative data from all the two programs and two areas cannot be combined in a grand model and then using area and program dummies as independent variables to test the differences directly rather than comparing the averages in an ad-hoc manner.

Response: Many thanks for this comment. We have now added this analysis as the supplementary table. Please see Table S3 and Table S4 in the supplementary file. We have also added the findings in the main text as follows “While comparing MTP utilization in an adjusted model including all schemes, we found that utilization of MTP was 33% lower among the MHI beneficiaries compared to HVS and 78% lower among the beneficiaries from Dhaka compared to the beneficiaries in Chattogram (supplementary table 3)” please see page 17

While comparing OOP in an adjusted model combining all schemes, overall OOP expenditure among the MHI beneficiaries were 41% higher compared to the beneficiaries of HVS. Furthermore, beneficiaries who lived in Dhaka spent 113% more OOP expenditure compared to the beneficiaries of Chattogram (supplementary table 4). Please see page 19

Comment 3. From the information provided in the tables, it appears that the surveyed households/individuals in Chittagong are quite different from surveyed units in Dhaka. It is important to mention possible reasons for the differences.

Response: Earlier study found that these two regions are more or less homogeneous based on demographic characteristics (Raheem et al. 2019). We observed difference in the household size, education level and other demographic characteristics except age (Table 4). These difference may be influenced by different types of schemes implemented in the two regions. In HVS scheme, member need not to pay for enrollment. Whereas in MHI scheme all members need to pay except ultra-poor. Therefore, it is expected that more poor people will enroll in MHI scheme compared to HVS scheme. We have considered differences in demographic characteristics in the multiple regression model while assessing the association. (Discussion; page 30; first para)

Comment 4. There appears to be a problem in the distribution of individuals by asset quintiles as presented in Table 4. I thought that the percent in each quintile has been defined separately by HVS Dhaka, HVS Chittagong and MHI Dhaka so that sum of column percentages will be 100%. That appears to be the case for HVS Dhaka and MHI Dhaka but not for HVS Chittagong. Please explain the reason for this discrepancy.

Response: Many thanks for this comment. Yes, this should be 100%. There was an typo in this section of table and we replaced 8.7% with correct value 38.7% in this version. Please see table 4.

---

## [Decision Letter · Decision Letter 1]

30 Jul 2021

Potential Effectiveness of Health Voucher Scheme and Micro-Health Insurance Scheme to Support the Poor and Extreme Poor in Selected Urban Areas of Bangladesh – An Assessment using a mixed-method approach

PONE-D-20-22715R1

Dear Dr. Ahmed,

We’re pleased to inform you that your manuscript has been judged scientifically suitable for publication and will be formally accepted for publication once it meets all outstanding technical requirements.

Kind regards,

M. Mahmud Khan

Academic Editor

PLOS ONE

Additional Editor Comments (optional):

I would suggest editing the title of the paper. The title could be: Effectiveness of Health Voucher and Micro-Health Insurance Schemes in Protecting the Health of Poor and Extreme Poor in Urban Bangladesh - An Assessment using a mixed-method approach

Reviewers' comments:

Reviewer's Responses to Questions

**Comments to the Author**

1. If the authors have adequately addressed your comments raised in a previous round of review and you feel that this manuscript is now acceptable for publication, you may indicate that here to bypass the “Comments to the Author” section, enter your conflict of interest statement in the “Confidential to Editor” section, and submit your "Accept" recommendation.

Reviewer #1: All comments have been addressed

2. Is the manuscript technically sound, and do the data support the conclusions?

Reviewer #1: Yes

3. Has the statistical analysis been performed appropriately and rigorously? 

Reviewer #1: Yes

4. Have the authors made all data underlying the findings in their manuscript fully available?

Reviewer #1: No

5. Is the manuscript presented in an intelligible fashion and written in standard English?

Reviewer #1: Yes

6. Review Comments to the Author

Reviewer #1: I thank the authors for addressing all the comments in the first round of review. The revised manuscript reads well, and I have no further major comments.

7. PLOS authors have the option to publish the peer review history of their article (what does this mean?). If published, this will include your full peer review and any attached files.

Reviewer #1: No

---

## [Editor Report · Acceptance letter]

19 Oct 2021

PONE-D-20-22715R1 

Effectiveness of Health Voucher Scheme and Micro-Health Insurance Scheme to Support the Poor and Extreme Poor in Selected Urban Areas of Bangladesh: An assessment using a mixed-method approach 

Dear Dr. Ahmed:

I'm pleased to inform you that your manuscript has been deemed suitable for publication in PLOS ONE. Congratulations! Your manuscript is now with our production department. 

Kind regards, 

on behalf of

Dr. M. Mahmud Khan 

Academic Editor

PLOS ONE